# OmniNet: A Multi-Modality Neural Network for Robust Remote Respiratory Rate Measurement from Facial Video

**Tsai-Ni Lin**[1,2]                                             TIFFANY10022000@GMAIL.COM

**An-Sheng Liu**[3]                                               D00921006@NTU.EDU.TW

**Li-Chen Fu**[*3]                                                 LICHEN@NTU.EDU.TW

[1] *Department of Mechanical Engineering, National Taiwan University, Taiwan*

[2] *Department of Bioengineering, Hong Kong Science and Technology University, Hong Kong*

[3] *Department of Electrical Engineering, National Taiwan University, Taiwan*

**Editors:** Accepted for publication at MIDL 2026

## Abstract

Remote respiratory rate (RR) measurement has gained traction in recent studies due to its ability to reduce healthcare professionals' workload and patient discomfort. Recent studies have targeted this problem through remote photoplethysmography (rPPG) to capture subtle facial color changes. However, this technique is sensitive to lighting and motion variations. To this end, we propose OmniNet, a multimodal neural network that integrates image data processed through 3D convolutional neural networks (3D CNNs) with point of interest (POI) motion data and passes the fused features to Bidirectional Long Short-Term Memory (BiLSTM) to model long-term temporal dependencies. OmniNet achieves state-of-the-art performance by effectively capturing comprehensive spatial and temporal information while reducing illumination variation and motion-induced artifacts. It also requires fewer computational resources and enables faster inference compared to Transformer networks. The code has been released on GitHub: https://github.com/tiffany-1002/OmniNet.

**Keywords:** Remote Respiratory Rate Measurement, Multimodal Learning

## 1. Introduction

The respiratory rate (RR) is a critical vital sign that offers predictive insight into a variety of health conditions, including obstructive sleep apnea, asthma, and other respiratory disorders. Traditional RR monitoring has relied on contact sensors (*e.g.* adhesive electrocardiogram (ECG) patches and respiration belts) that continuously record cardiac electrical activity or thoracic expansion. However, contact devices not only increase clinical workload but also cause discomfort or skin irritation during prolonged monitoring, especially in infants (Ethawi et al., 2018) and patients with dermatological conditions or burns.

Recent research in remote photoplethysmography (rPPG), a non-contact technique that detects subtle skin color variations through a RGB camera, has proposed various algorithms to obtain physiological signals. This method has been applied to estimate RR (Chen et al., 2019), heart rate (HR) (Malasinghe et al., 2022; Premkumar and Hemanth, 2022), heart rate variability (HRV) (Poh et al., 2011), and other physiological indicators. Although rPPG shows great potential, its reliance on skin color makes it sensitive to lighting variation (Tarassenko et al., 2014) and prone to motion artifacts caused by head movements (Qiu et al., 2022; Bousefsaf et al., 2013).

---

* Corresponding author

Pixel-based RR techniques offer an alternative approach to estimate RR by directly tracking periodic pixel motion induced by thoracoabdominal respiration, providing greater robustness under varying illumination conditions. Among the representative techniques, optical flow detects motion by computing flow vectors of moving regions over time (Koolen et al., 2015; Mateu-Mateus et al., 2020), while temporal differencing calculates pixel-wise intensity changes between consecutive video frames (Liu et al., 2020; Bai et al., 2010). These techniques are hampered by background noise in complex environments, often hindering the accurate separation of respiratory motion from unrelated movements (Bai et al., 2019). Careful selection of the region of interest (ROI) has also been proven to raise the quality of respiratory signal. Commonly selected ROIs include the chest, abdomen (Janssen et al., 2015), and facial areas such as the forehead, cheeks, or nose (Mehta and Sharma, 2020).

Deep learning (DL) methods have emerged as powerful techniques to measure physiological signals. Convolutional neural networks (CNNs) have shown strong capabilities both in image understanding (Girshick et al., 2014; Redmon et al., 2016) and extracting accurate meaningful rPPG signals from low-quality facial videos. While 2D CNNs focus on spatial information (Liu et al., 2020; Chen and McDuff, 2018), 3D CNNs effectively model both global spatial and temporal information across frames (Yu et al., 2019; Ghezzi et al., 2024).

Transformers, first used in Natural Language Processing (NLP) for sequence modeling, better capture long-term dependencies than convolutional networks and are well suited for modeling the periodic nature of respiratory signals due to their self-attention mechanism. Although Transformers achieve high accuracy, they suffer from relatively high complexity (Chen et al., 2024), and their temporal attention can be inaccurate, leading to phase shifts and irrelevant attention (Yu et al., 2022, 2023). In camera-based physiological measurement, where data are more limited than in other vision tasks, Transformers often perform worse than CNNs despite their strong modeling capabilities (Liu et al., 2023). Furthermore, Transformers require large-scale annotated facial videos to perform well, but such datasets are scarce and difficult to collect (Yue et al., 2023). CNNs are more effective in small-data settings but struggle to model long-term temporal dependencies.

Recurrent neural networks (RNNs) are widely used DL models well-suited for time series and sequential data. Long Short-Term Memory (LSTM) (Hochreiter and Schmidhuber, 1997) networks, a variant of RNNs, are effectively capture temporal dependencies and are often combined with CNNs for physiological signal estimation. Transformers require substantial training data, whereas LSTMs generally provide more efficient performance in environments with limited resources. While LSTMs are suitable for sequential modeling, their application to contactless RR estimation remains limited, as recent studies (Kumar et al., 2022; Rodrigues et al., 2024) have relied exclusively on contact-based datasets, without data derived from images or videos.

Moreover, existing contactless methods often rely on single-modality input and fail to integrate complementary information sources. To address these limitations, we propose a multimodal network named OmniNet. Several studies have explored multimodal approaches for respiratory or physiological signal estimation by combining facial motion, rPPG, or additional sensing modalities (Shao et al., 2025; Liao et al., 2024; Kong et al., 2024; Zheng, 2024; Gwak et al., 2024). These methods demonstrate the potential of multimodal fusion to improve robustness under noise and motion artifacts.

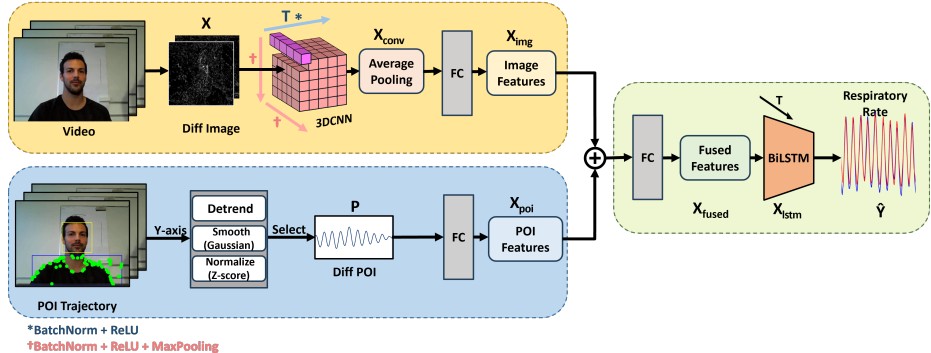

Figure 1: OmniNet framework.

By incorporating complementary multimodal inputs, our model reduces reliance on any single information source, which improves robustness to noise and leads to more stable respiratory rate estimation. OmniNet utilizes two types of information: frame differencing with a 3D CNN and POI motion trajectories. In this paper, we use a 3D CNN to capture temporal information, which is crucial for sequential signal extraction. Furthermore, the POI motion trajectories, inspired by OPOIRES (Deo Mehta and Sharma, 2023), is proposed to track meaningful points on the human torso rather than the fixed ROIs. Then we fuse these two types of features and feed them into a bidirectional LSTM (BiLSTM) to leverage its strength in modeling long-range temporal dependencies. We validate our approach on the COHFACE (Heusch et al., 2016) dataset against state-of-the-art methods, where it achieves superior performance. Moreover, the lightweight architecture of OmniNet ensures efficient deployment on mobile and embedded devices, emphasizing its relevance for practical real-world usage, especially in mobile healthcare and telemedicine.

## 2. Methodology

### 2.1. Overview of OmniNet Architecture

Illustrated in Figure 1, OmniNet is a multimodal DL model designed for robust respiratory signal estimation from facial videos under challenging conditions, *e.g.* illumination changes, head movements, and varying skin tones. Given a batch of input videos $\mathbf{X}$ and POI-based motion signals $\mathbf{P}$, OmniNet adopts a dual-branch architecture consisting of a video stream and a POI stream. The video stream processes $\mathbf{X}$ with a lightweight 3D CNN to extract motion-aware spatiotemporal features while preserving temporal resolution. These features are globally averaged across spatial dimensions and linearly projected to form the image embedding $\mathbf{X}_{\text{img}}$. In parallel, the POI stream passes $\mathbf{P}$ through a linear projection layer to obtain $\mathbf{X}_{\text{poi}}$. The two modality-specific features are concatenated along the last dimension and fused via a ReLU-activated fully connected layer, yielding $\mathbf{X}_{\text{fused}}$. Temporal modeling is then performed using a single-layer BiLSTM. The resulting sequence $\mathbf{X}_{\text{lstm}}$ is passed through a dropout layer and a final linear projection to estimate the respiratory signal $\hat{\mathbf{Y}}$, subsequently obtaining RR estimation via peak detection.

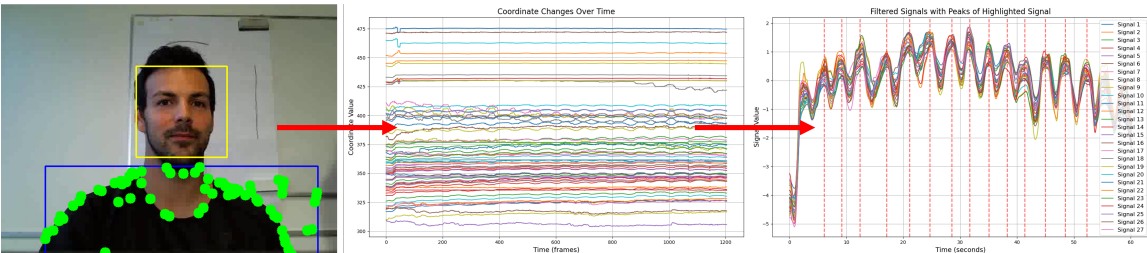

Figure 2: Visualization of POI selection: The system detects the face and defines an ROI below it, within which POIs are selected. The $y$-axis trajectories of these POIs are tracked over time, with each line representing the positional change of a single coordinate. Candidate signals are filtered after preprocessing, including detrending, smoothing, and normalization. Detected peaks are marked with red dashed lines indicate the estimated breathing cycles.

## 2.2. POI Selection

Following the previous literature (Deo Mehta and Sharma, 2023), we adopt the Viola-Jones algorithm for face detection due to its lightweight and real-time performance. As shown in Figure 2, after the face detection, a rectangular region below the face is defined to approximate the chest area, where respiratory motion is most prominent.

To enhance contrast, Contrast Limited Adaptive Histogram Equalization (CLAHE) and Gaussian smoothing are applied to the selected region. Harris corner detection is then performed to identify POIs, followed by sub-pixel refinement. A non-maximum suppression step with a minimum distance constraint enforces spatial diversity, retaining up to 30 points. These POIs are subsequently mapped to full-frame coordinates.

Only the $y$-axis displacement of each POI is preserved (Keall et al., 2006) after tracking POIs through frames using optical flow, resulting in one-dimensional temporal signals. Each signal is detrended, smoothed and standardized via z-score normalization:

$$\tilde{y}(t) = \frac{y(t) - \mu}{\sigma}, \tag{1}$$

where $\mu$ and $\sigma$ denote the mean and standard deviation of $y(t)$. The normalized autocorrelation function is then computed and scaled by its maximum to ensure consistency across signals, where $N$ denotes the length of the standardized signal $\tilde{y}(t)$:

$$R(\tau) = \sum_{t=\tau-N+1}^{N} \tilde{y}(t) \cdot \tilde{y}(t + N - \tau), \quad N \le \tau \le 2N - 1. \tag{2}$$

Signals with fewer than 2 or more than 30 autocorrelation peaks and with a mean difference of peak intervals exceeding 70 are discarded. To further enhance robustness, a Pearson correlation matrix $C \in \mathbb{R}^{S \times S}$ is computed among the autocorrelation functions of

$S$ candidate signals, and the mutual similarity score for each is defined as:

$$L_j = \sum_{i=1}^{S} C(i,j), \quad 1 \leq j \leq S. \tag{3}$$

The top $\lceil S/2 \rceil$ signals with the highest $L_j$ scores are retained as valid candidate signals.

## 2.3. Frame Differencing and Model Training

To extract RIM from video sequences, we employ a frame differencing strategy followed by a 3D CNN to capture both spatial and temporal dynamics of subtle respiratory movements.

**Frame Differencing** Given a facial video sequence $\mathbf{I}_1, \mathbf{I}_2, ..., \mathbf{I}_T$ of $T$ grayscale frames, we compute consecutive frame differences to highlight temporal motion: $\mathbf{X}_t = |\mathbf{I}_{t+1} - \mathbf{I}_t|$. This operation emphasizes motion features while reducing the impact of illumination changes. The resulting difference sequence $\mathbf{X}$ serves as an intermediate representation of potential respiratory activity. To preserve the temporal structure, we stack the differences as a new input volume to the subsequent 3D CNN encoder. Similarly for POI, we compute consecutive sample differences: $\mathbf{P}_t = \tilde{y}(t+1) - \tilde{y}(t)$. Afterward, we sum up the differences of different POIs and perform z-score normalization on the temporal dimension, producing a difference sequence $\mathbf{P}$, followed by the same process as the video stream.

**3D CNN Architecture** To process the stacked difference volume, we adopt a lightweight 3D CNN encoder to model spatial and short-term temporal patterns. Specifically, the architecture consists of three convolutional blocks, each of which contains a 3D convolution layer with kernel size $3 \times 3 \times 3$, followed by BatchNorm3D and ReLU activation. The first two blocks also has a 3D max pooling with kernel size $1 \times 2 \times 2$ to downsample spatial dimensions while preserving the temporal axis. The output feature maps are then averaged across the spatial dimensions and applied a linear projection to obtain $\mathbf{X}_{\text{img}}$ for temporal alignment with $\mathbf{X}_{\text{poi}}$ from the POI stream. This design allows the model to capture fine-grained temporal cues from motion-only input, making it more robust to appearance variations, head pose changes, and illumination artifacts compared to directly analyzing RGB frames.

**BiLSTM Architecture** Following the fusion of image and POI features, we apply a single-layer BiLSTM. This allows the model to incorporate both past and future contexts for each time step. A dropout layer is applied to the LSTM output before a final fully connected layer maps each temporal feature vector to a scalar prediction $\hat{Y}(t)$.

**Optimization Strategy** The loss function is the mean squared error (MSE) between the predicted respiration belt signals $\hat{Y}(t)$ and the ground truth values $Y(t)$, defined as:

$$\mathcal{L}_{\text{MSE}} = \frac{1}{T} \sum_{t=1}^{T} \left( \hat{Y}(t) - Y(t) \right)^2, \tag{4}$$

where $T$ denotes the number of temporal samples in each sequence. We refer the reader to Appendix A for more details.

## 2.4. Peak Detection and RR Estimation

To estimate the RR, we first preprocess the respiration belt signal using Savitzky-Golay smoothing, followed by a fourth-order Butterworth band-pass filter with cutoff frequencies of 0.1–0.4 Hz to suppress noise and baseline drift. Peak detection is performed using the `find_peaks` signal processing functions build-in in the SciPy library, where a dynamic threshold is applied to exclude spurious local maxima. The minimum peak distance is set to 6 frames to avoid detecting peaks caused by motion artifacts.

If multiple valid peaks are found, we compute the pairwise inter-peak intervals and calculate the average breathing cycle. The RR (in breaths per minute) is then given by: $RR = 60/\bar{\Delta}$, where $\bar{\Delta}$ denotes the average peak interval in seconds. If fewer than two peaks are detected, the signal is excluded from the RR computation.

## 2.5. Further Discussion on Multimodal Fusion

Although both modalities are derived from the same input video, they encode fundamentally different information at the representation level. The POI branch captures sparse, low-dimensional geometric motion by tracking selected landmarks, whereas the video branch models dense, high-dimensional appearance and pixel-level temporal motion patterns using 3D convolution. Consequently, the two modalities exhibit distinct inductive biases and capture complementary rather than redundant aspects of respiratory motion.

From a machine learning perspective, features from one modality could indeed dominate or interfere with the other if early fusion or shared representations were used. To address this risk, the two branches are processed by independent encoders without parameter sharing, which prevents feature leakage and enforces modality disentanglement during representation learning. Unimodal baselines are trained independently using only their respective inputs, ensuring that their performance reflects the intrinsic properties of each modality alone.

In the multimodal setting, fusion is performed using a late fusion strategy. Specifically, modality-specific features are concatenated and projected through a lightweight learnable linear layer before temporal modeling. This design ensures that fusion occurs at the feature level rather than at the raw signal level, limits cross-modal interference, and allows the network to implicitly adjust the relative contribution of each modality based on data reliability. Empirically, this approach yields consistent improvements over both single-modality variants, confirming that fusion leverages complementary cues instead of amplifying shared noise.

## 3. Experiments

### 3.1. Dataset and Experimental Setup

Due to the limited availability of RR datasets, we rely solely on the COHFACE (Heusch et al., 2016) dataset for our experiments.

**COHFACE:** The COHFACE dataset contains 160 one-minute RGB videos of 40 subjects (12 women, 28 men), with each subject contributing two videos under studio lighting and two under natural light. Videos were recorded at $640 \times 480$ resolution and 20 Hz, while respiration belt and blood volume pulse signals from the same subject were simultaneously recorded at 256 Hz. Official experimental splits are provided under three protocols: all,

clean, and natural. Each protocol offers standard train/dev/test partitions for evaluating model robustness across varied illumination conditions. Data from the same subject will not present in multiple partitions at the same time, preventing temporal and subject leakage.

The original 20 Hz video is temporally downsampled to 4 Hz by selecting one frame every five frames. This method aligns the visual input with the respiration belt signal, which is also downsampled to 4 Hz, enabling a more efficient calculation and facilitating synchronization between modalities. We set the input window size to 240 frames, which corresponds to the one-minute length of both the differential image sequences and the respiration belt readings.

The predicted respiration belt signals and the ground truth values are processed by the same procedures described in Section 2.4, obtaining the estimated and real RR. We evaluated the performance of the model using three standard metrics: mean absolute error (MAE), root mean square error (RMSE), and Pearson correlation coefficient (PCC). Standard deviation of absolute error (STD) is also used when comparing with OPOIRES (Deo Mehta and Sharma, 2023). These metrics are described in Appendix B.

In addition, a detailed description about the experimental environment is provided in Appendix C.

## 3.2. Intra-Database Tests

Table 1: Performance comparison between existing methods and our method on the CO-HFACE dataset.

| Methods | MAE (bpm) ↓ | RMSE (bpm) ↓ | PCC ↑ |
|---|---|---|---|
| DeepPhys (Chen and McDuff, 2018) | 3.21 | 6.56 | 0.52 |
| TS-CAN (Liu et al., 2020) | 2.97 | 5.49 | 0.63 |
| TS-DAN (Ren et al., 2021) | 2.83 | 5.72 | 0.59 |
| PhysNet (Yu et al., 2019) | 2.31 | 4.77 | 0.67 |
| EVM-MPP (Alnaggar et al., 2023) | 1.63 | 2.10 | 0.45 |
| PhysFormer (Yu et al., 2022) | 1.44 | 2.29 | 0.62 |
| ACTNet (Chen et al., 2024) | 1.08 | 1.57 | 0.81 |
| CliffPhys (Ghezzi et al., 2024) | 0.83 | 1.97 | 0.86 |
| OmniNet | **0.24** | **0.42** | **0.99** |

All results are reported on the COHFACE test set. The CliffPhys model is pre-trained on the SCAMPS dataset and fine-tuned using the COHFACE training set. All other results, including TS-CAN, TS-DAN, PhysFormer, and ACTNet, are cited from ACTNet.

The results on the COHFACE dataset are shown in Table 1. We follow the official training and testing protocols provided by the dataset. OmniNet achieves the lowest MAE (0.24) and RMSE (0.42) and the highest PCC (0.99) on the COHFACE test set. These results surpass all compared baselines, demonstrating the effectiveness of our framework for accurate RR estimation.

Table 2: Performance comparison of our method on the COHFACE dataset.

| Train Set | Test Set | MAE (bpm) ↓ | RMSE (bpm) ↓ | PCC ↑ |
|-----------|----------|-------------|--------------|-------|
| All | All | 0.242 | 0.421 | 0.991 |
| All | Clean | **0.223** | **0.404** | **0.993** |
| All | Natural | 0.261 | 0.437 | 0.991 |
| Clean | Clean | 0.234 | 0.511 | 0.989 |
| Natural | Natural | 0.307 | 0.588 | 0.985 |

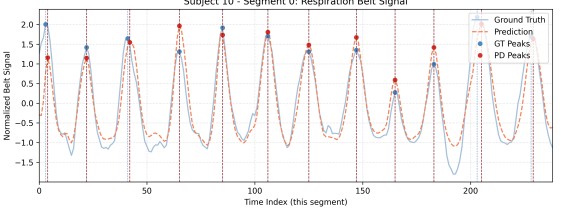 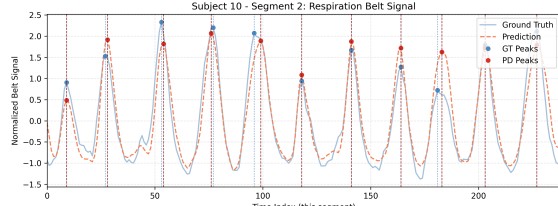

(a) Clean lighting condition      (b) Natural lighting condition

Figure 3: Comparison of predicted and ground truth respiration belt signals for Subject 10 using the model trained on the whole COHFACE dataset. Dots indicate peak positions of the ground truth and predicted signals.

We further assess illumination robustness by training and testing OmniNet on the clean and natural subsets separately, reporting all results to three-decimal precision. As shown in Table 2, training on the full dataset (All) and testing on the clean subset achieves the lowest MAE (0.223) and RMSE (0.404) and the highest PCC (0.993). Although the clean subset yields slightly better performance, the differences across illumination conditions remain small, indicating that OmniNet is robust to lighting variations and performs well even under natural lighting.

Moreover, training on the whole dataset consistently outperforms training solely on a single subset, likely due to the increased diversity and volume of training data that improve the model's generalizability. For example, the MAE reduces from 0.307 to 0.261 and the RMSE from 0.588 to 0.437 with the PCC increasing from 0.985 to 0.991 when switching from natural-subset training and testing to whole-set training and natural-subset testing. The consistent trend across MAE, RMSE, and PCC reinforces the effectiveness of using more comprehensive training data.

Figure 3 illustrates the respiration belt signals predicted by models trained on the whole dataset for a representative sample under clean and natural lighting conditions. The predicted curves closely follow the ground truth in the clean condition, with accurate peak detection, while predictions under natural lighting exhibit slight deviations. To further analyze the effect of training data quality, Figure 4 compares the results when training and

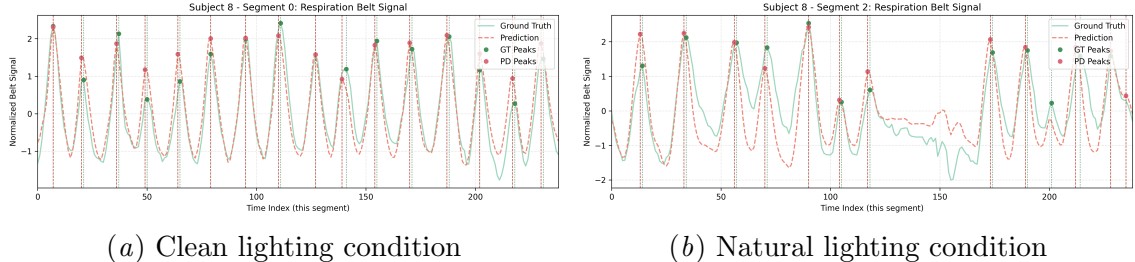

(*a*) Clean lighting condition          (*b*) Natural lighting condition

Figure 4: Comparison of predicted and ground truth respiration belt signals for Subject 8 using models trained and tested on the same condition of the COHFACE dataset under different lighting settings. Dots indicate peak positions of the ground truth and predicted signals.

testing solely on clean versus natural subsets, using a representative sample with notably divergent behavior. When trained on the natural subset, the model produces significant mismatches, including unstable fluctuations and incorrect peak detection. In contrast, training on the clean subset results in more accurate and stable predictions.

Since our method was inspired by the framework proposed in OPOIRES (Deo Mehta and Sharma, 2023), we conducted a direct comparison using their reported best setting (60-second input). For a fair comparison, we also adopted STD as one of the evaluation metrics, in addition to MAE and RMSE. All models are trained on the whole dataset and evaluated separately on the clean and natural test subsets.

Table 3: Performance comparison between OPOIRES and our method on the COHFACE two test set, using models trained on the whole dataset.

| Test Set | Methods | MAE (bpm) ↓ | RMSE (bpm) ↓ | STD (bpm) ↓ |
|---|---|---|---|---|
| Clean | OPOIRES (60s) (Deo Mehta and Sharma, 2023) | 0.48 | 1.13 | 1.03 |
| | OmniNet | **0.22** | **0.40** | **0.34** |
| Natural | OPOIRES (60s) (Deo Mehta and Sharma, 2023) | 0.60 | 1.40 | 1.28 |
| | OmniNet | **0.26** | **0.44** | **0.35** |

As shown in Table 3, our method significantly outperforms OPOIRES. These results demonstrate the robustness and effectiveness of our multimodal design. Compared to relying solely on POI selection, incorporating multiple modalities leads to more accurate and stable RR estimation.

To evaluate whether the proposed method satisfies commonly adopted clinical agreement criteria ($\pm2$ breaths per minute relative to manual counting as the gold standard, as reported in Goldfine et al. (2024)), we compute the absolute RR error using the model trained on the full dataset. Detailed results are reported in Appendix D. We observe that all predictions

exhibit absolute errors of approximately 1 bpm or less, indicating that OmniNet meets established clinical accuracy requirements.

### 3.3. Ablation Studies

3.3.1. Impact of 3D CNN

Table 4: Performance comparison under ablation settings on the COHFACE dataset, using models trained on the whole dataset.

| Test Set | Methods | MAE (bpm) ↓ | RMSE (bpm) ↓ | PCC ↑ |
|---|---|---|---|---|
| All | OmniNet | **0.242** | **0.421** | **0.991** |
| | OmniNet (*w/o 3D CNN*) | 0.269 | 0.500 | 0.988 |
| Clean | OmniNet | **0.223** | **0.404** | **0.993** |
| | OmniNet (*w/o 3D CNN*) | 0.273 | 0.552 | 0.986 |
| Natural | OmniNet | **0.261** | **0.437** | **0.991** |
| | OmniNet (*w/o 3D CNN*) | 0.265 | 0.442 | 0.991 |

In this ablation study, we evaluated the impact of removing the 3D CNN module. As shown in the first two rows of Table 4, removing the 3D CNN results in only a slight performance drop on the whole dataset (All).

On both the natural and clean test sets, the full model consistently outperforms the model without 3D CNN. However, the changes are relatively small, indicating that the POI branch alone can still capture meaningful temporal features. These findings highlight the value of our multimodal architecture. While the POI branch offers robustness in noisy conditions, the integration of image-level features through 3D CNN provides complementary information that is especially beneficial in clean and stable environments. A detailed comparison of all POI-related hyperparameters is provided in Appendix E.

3.3.2. Impact of POI Selection

For all these ablation experiments, all CNN-related hyperparameters are fixed to the best-performing configuration to ensure a fair comparison. As shown in Table 5, removing the POI modality leads to substantial performance degradation, causing much greater impact compared to removing 3D CNN. This may be because respiratory motion is highly localized and periodic, making POI features more effective in capturing relevant temporal patterns. In contrast, 3D CNN processes broader spatiotemporal regions, which are more susceptible to noise from lighting variations or non-respiratory motion. Nonetheless, even when either image or POI modality is ablated, OmniNet still outperforms all baselines. These results highlight the effectiveness and robustness of the model's multimodal design, as well as the complementary nature of image and POI features.

Table 5: Performance comparison under ablation settings on the COHFACE dataset, using models trained on the whole dataset.

| Test Set | Methods | MAE (bpm) ↓ | RMSE (bpm) ↓ | PCC ↑ |
|---|---|---|---|---|
| All | OmniNet | **0.242** | **0.421** | **0.991** |
| | OmniNet (*w/o POI*) | 0.744 | 1.191 | 0.939 |
| Clean | OmniNet | **0.223** | **0.404** | **0.993** |
| | OmniNet (*w/o POI*) | 0.707 | 1.126 | 0.949 |
| Natural | OmniNet | **0.261** | **0.437** | **0.991** |
| | OmniNet (*w/o POI*) | 0.782 | 1.252 | 0.931 |

We further assess robustness under partial information loss by conducting test-time degradation experiments, where the model is trained with both modalities but evaluated with one modality deliberately corrupted while the other remains unchanged. Under severe single-modality corruption, the model maintains strong performance across all metrics, exhibiting only moderate degradation relative to the original setting. Detailed results are reported in Appendix F.

To assess whether the reported performance gains are statistically meaningful, we conducted subject-level statistical validation using paired Wilcoxon signed-rank tests, detailed in Appendix G.

### 3.4. Analysis under Facial Occlusion Scenarios

While the robustness evaluation scenarios provided by the COHFACE dataset mainly focus on varied illumination, *i.e.* clean vs. natural lighting conditions, real-world deployment for remote RR measurement also involves additional challenges such as facial occlusion, head movement, speaking, and camera motion. To evaluate the effect of facial occlusion, we have added additional robustness analysis by simulating this factor in Appendix H. The results show that respiration estimation performance remains consistent across different occlusion strategies. Although this simulation does not exhaustively cover all real-world occlusion patterns, it provides a meaningful evaluation of robustness under partial facial occlusion. Systematic evaluation of these factors, however, requires reliable ground-truth respiration signals, which are not available in COHFACE. We therefore avoid over-claiming robustness and identify broader stress testing as an important direction for future work.

### 3.5. Complexity Analysis

We also compared the complexity of the proposed method with the baselines, from which we can conclude that OmniNet achieved state of the art not only in performance but also in complexity. We refer the reader to Appendix I for more details.

### 3.6. Comparison Between CNN- and Transformer-Based Encoders

Although OmniNet is a multimodal framework, its architecture is modular, allowing the image-based encoder to be replaced without altering the overall design. To account for the growing use of Transformers in video modeling, we conduct a controlled comparison by replacing the 3D CNN with a lightweight Vision Transformer (ViT) while keeping all other components unchanged.

As shown in Appendix J, CNN-based encoders achieve better performance under limited-data regimes. A more detailed analysis and discussion are provided in the appendix.

## 4. Conclusion and future work

In this paper, we proposed OmniNet, a lightweight multimodal framework for remote respiratory rate (RR) estimation. Our method integrates pixel-based POI selection with a 3D CNN module using frame differencing and a BiLSTM decoder to model temporal dynamics. Through extensive experiments on the COHFACE dataset, OmniNet consistently outperforms existing methods under all conditions, particularly in clean data scenarios. Notably, our network has the smallest parameter size and computational complexity among all compared approaches, making it suitable for deployment on portable medical devices.

Despite its strong performance, our method has limitations. RR is a semi-voluntary signal that can be consciously controlled by subjects, potentially introducing noise in measurements. Furthermore, the current pipeline relies on motion cues extracted after face detection, with respiration primarily inferred from the lower face, neck, and shoulder regions. Consequently, large pose variations, significant body motion, speech activity, and scenarios involving complete facial coverage may degrade performance by altering or obscuring respiration-related motion patterns.

Although explicit measures are taken to prevent temporal and subject leakage through non-overlapping temporal windows and subject-independent data splits, the study is conducted on a relatively small, single-dataset benchmark. Therefore, residual intra-subject correlations related to recording conditions may persist and limit generalization to unseen acquisition settings.

Nevertheless, we believe that OmniNet remains valuable in specific clinical contexts. It is particularly suitable for populations with limited voluntary motion, such as burn patients, newborns, or patients during sleep, where contact-based measurements are impractical. The proposed approach is also well suited for telemedicine and home-care scenarios, enabling unobtrusive monitoring of respiratory trends during sleep.

In future work, we plan to 1) further validate the generalizability of OmniNet through cross-dataset evaluation on additional RR benchmarks, 2) improve the pipeline by developing end-to-end architectures, and 3) extend the framework to comprehensive vital sign estimation in practical clinical environments.

## Acknowledgments

This research was supported by the National Science and Technology Council & National Taiwan University, Taiwan, under the grant numbers 113-2634-F-002-002- , 113-2223-E-002-006- & 113-2221-E-002-127-MY3.

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

## Appendix A. Optimization Strategy

The model is trained with a batch size of 8. We use the Adam optimizer, and the learning rate is tuned per model to maximize performance, with $5 \times 10^{-3}$ serving as the common starting point in most configurations. To stabilize training and prevent gradient explosion at the early stage, we adopt a linear learning rate schedule with warm-up. Specifically, the learning rate increases linearly during the first $N_{\text{warmup}} = 10$ steps, followed by a linear decay until the total number of training steps $N_{\text{total}} = 100$. The learning rate at step $i$ is computed as $\eta_i \cdot \text{lr}_{\text{init}}$, where

$$\eta_i = \begin{cases} \frac{i}{N_{\text{warmup}}} & \text{if } i < N_{\text{warmup}}, \\ \frac{N_{\text{total}} - i}{N_{\text{total}} - N_{\text{warmup}}} & \text{otherwise.} \end{cases} \tag{5}$$

This scheduling strategy facilitates stable convergence by allowing sufficient exploration in the initial phase and finer adjustment in the later training stages. Early stopping is triggered after 10 epochs of no improvement. At each epoch, we save the best model based on the loss of the development set, which is used for all subsequent evaluations.

## Appendix B. Evaluation Metrics

The four evaluation metrics are described as follows, where $n$ is the number of samples, $\hat{r}_i$ denotes the predicted RR, and $r_i$ is the ground truth value for the $i$-th subject.

**Mean absolute error (MAE)** measures the average deviation between predicted and actual RR (in bpm):

$$\text{MAE} = \frac{1}{n} \sum_{i=1}^{n} |\hat{r}_i - r_i|. \tag{6}$$

**Root mean square error (RMSE)** quantifies overall prediction error magnitude:

$$\text{RMSE} = \sqrt{\frac{1}{n} \sum_{i=1}^{n} (\hat{r}_i - r_i)^2}. \tag{7}$$

**Pearson correlation coefficient (PCC)** indicates linear correlation between predicted and true values:

$$\text{PCC} = \frac{\sum_{i=1}^{n} (\hat{r}_i - \bar{\hat{r}})(r_i - \bar{r})}{\sqrt{\sum_{i=1}^{n} (\hat{r}_i - \bar{\hat{r}})^2} \cdot \sqrt{\sum_{i=1}^{n} (r_i - \bar{r})^2}}. \tag{8}$$

**Standard deviation of absolute error (STD)** measures the variability of the absolute errors:

$$\text{STD} = \sqrt{\frac{1}{n} \sum_{i=1}^{n} \left( |\hat{r}_i - r_i| - \frac{1}{n} \sum_{j=1}^{n} |\hat{r}_j - r_j| \right)^2}. \tag{9}$$

## Appendix C. Experimental Environment

The experimental environment consists of a Windows 11 64-bit operating system running on a machine equipped with an Intel Core i9-14900HX processor (2.20 GHz) and 32 GB of system memory. All experiments were conducted using an NVIDIA GeForce RTX 4060 GPU with CUDA version 12.5. The proposed method is implemented in Python (version 3.12) using the PyTorch (version 2.6) deep learning framework.

## Appendix D. Clinical Agreement Analysis of Absolute Error Distributions

To visualize the error distributions, we adopt the standard Tukey boxplot convention, where the box represents the interquartile range (IQR) between the first quartile (Q1, 25th percentile) and the third quartile (Q3, 75th percentile), the central line denotes the median, and the whiskers extend to values within $1.5 \times$ IQR. Values beyond this range are treated as outliers.

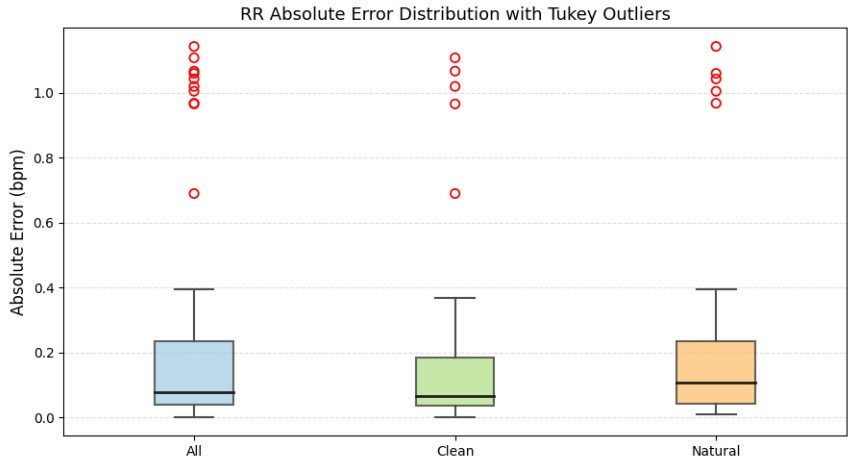

Figure 5: Absolute respiratory rate error distributions across different test sets ($N = 64$, 32, 32) using the model trained on the whole dataset.

As shown in Figure 5, all predictions fall well within $\pm 1$ bpm, and at least 75% of the samples exhibit absolute errors below 0.233 bpm, indicating strong estimation consistency. These results confirm that the proposed approach achieves clinically acceptable accuracy and demonstrates strong potential for practical deployment in real-world respiratory monitoring scenarios.

## Appendix E. Analyses of POI Selection Parameters

We tuned the number of POIs (#POIs), quality level, and minimum distance (MD) in the POI stream, showing the results in Table 6. Based on these analyses, we fix the POI-related

Table 6: Analyses of POI-related hyperparameters on RR estimation performance on the whole COHFACE dataset.

| #POIs | Quality level | MD | MAE (bpm) ↓ | RMSE (bpm) ↓ | PCC ↑ |
|-------|---------------|----|-----|------|-----|
| *Default configuration* | | | | | |
| 100 | $10^{-4}$ | 2 | **0.242** | **0.421** | **0.991** |
| *Varying number of POIs, others fixed to default* | | | | | |
| 50 | $10^{-4}$ | 2 | 0.272 | 0.503 | 0.988 |
| 150 | $10^{-4}$ | 2 | 0.255 | 0.518 | 0.987 |
| *Varying quality level, others fixed to default* | | | | | |
| 100 | $10^{-2}$ | 2 | 0.355 | 0.723 | 0.976 |
| 100 | $10^{-3}$ | 2 | 0.248 | 0.476 | 0.989 |
| 100 | $10^{-5}$ | 2 | 0.264 | 0.523 | 0.987 |
| *Varying minimum distance (MD), others fixed to default* | | | | | |
| 100 | $10^{-4}$ | 4 | 0.277 | 0.512 | 0.987 |
| 100 | $10^{-4}$ | 6 | 0.273 | 0.575 | 0.984 |

hyperparameters to their best-performing values and adopt this configuration as the default in all subsequent experiments.

## Appendix F. Test-Time Robustness Analysis under Single-Modality Degradation

To assess robustness under partial information loss, we conduct additional test-time degradation experiments. The model is trained using both modalities on the whole dataset and evaluated under settings where one modality is deliberately corrupted while the other remains unchanged, simulating partial modality failure at inference time. Additive Gaussian noise is adopted as the corruption type for both modalities. Image noise is injected directly in the 8-bit `uint8` intensity space, whereas POI noise is applied in the z-score normalized coordinate space. Therefore, noise magnitudes are not directly comparable across modalities.

For POI trajectories, respiration-induced displacements in the normalized space typically lie within a small range around zero, and we therefore apply a relatively large noise level ($\sigma = 1$) to simulate severe degradation of the POI modality. For image corruption, we empirically observe that small noise levels (*e.g.* $\sigma \leq 5$ ) have negligible impact. Consequently, we adopt a stronger noise level corresponding to a peak signal-to-noise ratio

(PSNR) of approximately 18.6 dB. According to Tian et al. (2023), PSNR values below 20 dB correspond to unacceptable image quality.

Table 7: Performance under test-time single-modality degradation.

| Degradation Setting | MAE (bpm) ↓ | RMSE (bpm) ↓ | PCC ↑ |
|---|---|---|---|
| Original | **0.242** | **0.421** | **0.991** |
| POI noise ($\sigma = 1$, z-score space) | 0.379 | 0.717 | 0.976 |
| Image noise ($\sigma = 30$ 8-bit space) | 0.294 | 0.524 | 0.987 |

Table 7 summarizes the performance under different test-time degradation settings. Even under severe single-modality degradation, the model maintains strong performance across all evaluation metrics. These results indicate that the proposed multimodal framework leverages complementary information across modalities and achieves robust performance without strict dependence on any single modality.

## Appendix G. Statistical Validation of Ablation Studies

Table 8: Subject-level statistical validation of ablation studies on the COHFACE dataset. $\Delta$MAE denotes the increase in MAE after removing a component (ablated vs. full). Statistical significance is evaluated using paired Wilcoxon signed-rank tests with bootstrap 95% confidence intervals.

| Condition | Comparison | N | $\Delta$MAE (bpm) | 95% CI | $p$-value | Cohen's $d_z$ |
|---|---|---|---|---|---|---|
| All | Full vs. w/o 3D CNN | 16 | 0.027 | $[-0.027,\ 0.087]$ | 0.221 | 0.22 |
| Clean | Full vs. w/o 3D CNN | 16 | 0.050 | $[-0.057,\ 0.170]$ | 0.248 | 0.20 |
| Natural | Full vs. w/o 3D CNN | 16 | 0.004 | $[-0.003,\ 0.012]$ | 0.249 | 0.24 |
| All | Full vs. w/o POI | 16 | 0.502 | $[0.237,\ 0.875]$ | **0.00021** | 0.73 |
| Clean | Full vs. w/o POI | 16 | 0.484 | $[0.197,\ 0.898]$ | **0.00076** | 0.64 |
| Natural | Full vs. w/o POI | 16 | 0.521 | $[0.199,\ 0.906]$ | **0.00269** | 0.70 |

To assess whether the reported performance gains are statistically meaningful, we conducted subject-level statistical validation using paired Wilcoxon signed-rank tests. All comparisons were performed on a per-subject basis to avoid bias arising from correlated samples. In addition, we report bootstrap-based 95% confidence intervals and effect sizes (Cohen's $d_z$) to quantify the magnitude of the observed differences. To estimate uncertainty, we further computed 95% confidence intervals of the mean MAE difference (ablated vs. full) via non-parametric bootstrap resampling (20,000 iterations with a fixed random seed for reproducibility). All reported statistics are therefore based on subject-level paired measurements, ensuring a fair and statistically sound comparison.

The statistical results in Table 8 show that removing the POI branch leads to a statistically significant increase in MAE across the all, clean, and natural conditions ($\Delta$MAE $\approx$ 0.48–0.52 bpm, $p < 0.01$, with medium-to-large effect sizes), confirming that the POI branch is a major contributor to the observed performance improvements. In contrast, removing the 3D CNN results in only small and statistically non-significant changes in MAE across all conditions ($p > 0.22$, small effect sizes), indicating a limited but consistent auxiliary contribution.

## Appendix H. Detailed Analysis under Facial Occlusion Scenarios

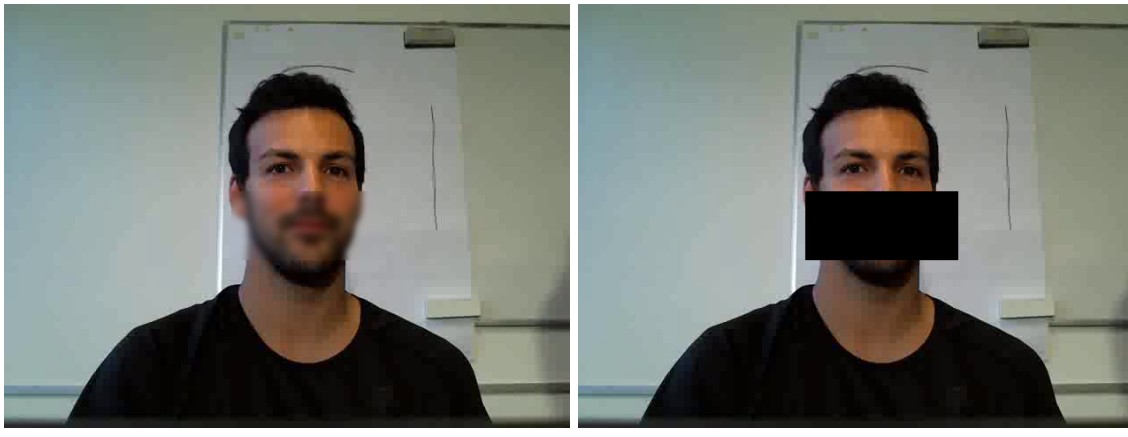

($a$) blur-based occlusion        ($b$) block-based occlusion

Figure 6: Qualitative illustration of facial occlusion modeling using blur-based and block-based occlusion on the lower facial region to simulate occlusion-related noise, such as mask wearing or hand occlusion.

Given the limited number of available public datasets, we conduct additional simulations on the COHFACE dataset to evaluate the robustness of our method under realistic noise conditions. We focus on facial occlusions commonly encountered in practice, such as mask wearing or partial hand occlusion, which may affect face detection and motion extraction.

We simulate lower-face occlusions using two strategies: blur-based and block-based masking, as illustrated in Figure 6. The occlusion is applied only to the lower facial region, while the eye region is preserved, as this setting reflects common real-world occlusion patterns and maintains reliable face detection.

Our framework is modular and detector-agnostic, allowing different face detectors to be used without modifying the core model. By default, we adopt the Viola-Jones detector for efficiency, while a MediaPipe-based detector is optionally evaluated in occlusion-heavy scenarios to ensure stable face localization. This choice introduces only a modest increase in model complexity (0.135M parameters).

The results in Table 9 show that the overall respiration estimation performance remains highly consistent across different occlusion modeling strategies. Although this simulation

Table 9: Performance comparison under different occlusion strategies using the OmniNet model, trained on the whole dataset.

| Test Set | Strategy | MAE (bpm) ↓ | RMSE (bpm) ↓ | PCC ↑ |
|----------|----------|-------------|--------------|-------|
| All | Original | **0.242** | **0.421** | **0.991** |
| | Blur | 0.272 | 0.507 | 0.988 |
| | Block | 0.264 | 0.526 | 0.987 |
| Clean | Original | **0.223** | **0.404** | **0.993** |
| | Blur | 0.265 | 0.538 | 0.987 |
| | Block | 0.244 | 0.517 | 0.988 |
| Natural | Original | **0.261** | **0.437** | **0.991** |
| | Blur | 0.278 | 0.473 | 0.989 |
| | Block | 0.284 | 0.534 | 0.986 |

does not exhaustively cover all possible real-world occlusion patterns, it provides a meaningful evaluation of the model's robustness. In OmniNet, the POI-based motion representation places greater emphasis on shoulder and upper-torso motion cues, which are less sensitive to facial visibility. Consequently, performance differences across settings remain limited.

## Appendix I. Complexity Analysis

Table 10: Comparison of the complexity between existing methods and our method.

| Methods | Param. (M) ↓ | MACs (G) ↓ | MAE (bpm) ↓ |
|---------|--------------|------------|-------------|
| DeepPhys (Chen and McDuff, 2018) | 7.50 | 111.76 | 3.21 |
| TS-CAN (Liu et al., 2020) | 7.50 | 111.76 | 2.97 |
| TS-DAN (Ren et al., 2021) | 7.50 | 111.91 | 2.83 |
| PhysNet (Yu et al., 2019) | 0.73 | 65.19 | 2.31 |
| PhysFormer (Yu et al., 2022) | 7.03 | 47.01 | 1.44 |
| ACTNet (Chen et al., 2024) | 20.72 | 77.84 | 1.08 |
| OmniNet | **0.20** | **15.01** | **0.24** |

All results, including TS-CAN, TS-DAN, PhysFormer, and ACTNet, are cited from ACTNet.

The number of parameters (Param.) and multiply-accumulate operations (MACs) are shown in Table 10. The parameters in the proposed method consist of two parts: the pretrained models via AdaBoost in the Viola-Jones algorithm and the OmniNet, including 90K and 114K parameters, respectively. We omitted the MACs of the Viola-Jones and subse-

quent algorithms, as they are negligible compared to those of OmniNet, whose complexities are measured by `thop`. We can observe that OmniNet achieved the state of the art in both complexity and performance, demonstrating the excellence of the method.

## Appendix J. Detailed Comparison Between CNN- and Transformer-Based Encoders

Recent work by Liu et al. (2023) systematically compared CNN- and Transformer-based models for camera-based vital sign measurement. Their results show that Transformer-based models require substantial optimization and large-scale pretraining to outperform even relatively shallow CNNs, which is difficult to achieve in physiological video analysis due to limited data availability.

Table 11: Comparison between 3D CNN and ViT encoders trained on the whole COHFACE dataset under different testing conditions.

| Test Set | Encoder | MAE (bpm) ↓ | RMSE (bpm) ↓ | PCC ↑ |
|---|---|---|---|---|
| All | 3D CNN | **0.242** | **0.421** | **0.991** |
| | ViT | 0.280 | 0.526 | 0.986 |
| Clean | 3D CNN | **0.223** | **0.404** | **0.993** |
| | ViT | 0.258 | 0.459 | 0.991 |
| Natural | 3D CNN | **0.261** | **0.437** | **0.991** |
| | ViT | 0.302 | 0.586 | 0.985 |

Table 12: Comparison between 3D CNN and ViT encoders under ablation settings on the COHFACE dataset, using models trained on the whole dataset.

| Test Set | Encoder | MAE (bpm) ↓ | RMSE (bpm) ↓ | PCC ↑ |
|---|---|---|---|---|
| All | 3D CNN (*w/o POI*) | **0.744** | **1.191** | **0.939** |
| | ViT (*w/o POI*) | 1.824 | 2.581 | 0.664 |
| Clean | 3D CNN (*w/o POI*) | **0.707** | **1.126** | **0.949** |
| | ViT (*w/o POI*) | 1.612 | 2.293 | 0.716 |
| Natural | 3D CNN (*w/o POI*) | **0.782** | **1.252** | **0.931** |
| | ViT (*w/o POI*) | 2.036 | 2.840 | 0.621 |

Motivated by these observations, we conduct a controlled comparison by replacing the 3D CNN with a lightweight Vision Transformer (ViT) (0.91M parameters), and report the results in Table 11 and Table 12. These results indicate that under relatively small data regimes, CNNs exhibit stronger capability in effectively supporting the overall model. The observed performance gains are primarily driven by the quality of the POI-based motion representation rather than the specific choice of image-based encoder. Once respiration-relevant motion trajectories are explicitly extracted, even lightweight image models achieve near-saturated performance. Although CNN-based and Transformer-based models achieve similar accuracy on the controlled benchmark, the CNN is more data-efficient and robust under reduced training data and corrupted POI conditions, with substantially fewer parameters.

