# OpenReview forum: "OmniNet: A multi-modality neural network for robust remote respiratory rate measurement from facial video"
_MIDL.io/2026/Conference — MIDL 2026 Poster_

### Official Review · Reviewer_yEyL · 2026-01-08

**Confidence:** 4
**Preliminary Rating:** 2
**Final Rating:** 3

**Summary:**

The authors aim to develop a robust, accurate, and computationally efficient method for remote respiratory rate estimation from facial cues in videos, thus addressing key limitations of existing camera-based physiological monitoring approaches. Specifically, the authors developed model aims to improve robustness against edge and challenging cases such as illumination variation and motion artifacts, which are commonly encountered in real-world deployment. The authors also focus on maintaining real-time processing for resource-constrained deployment scenarios.

**Strengths:**

The paper's motivation about real-time deployment and robustness to illumination changes and motion artifacts is a timely problem, especially when we are seeing a surge in remote and home monitoring systems for healthcare applications. The proposed network has the potential to be integrated into practice for real-time deployment especially for privacy-sensitive applications, where the data could not be uploaded to cloud servers and instead needs to be processed locally on a machine. Below are the major strengths:

1. Clear motivation and practical relevance - The paper addresses an important and clinically meaningful problem: non-contact respiratory rate estimation from facial video. The paper motivation is well written, particularly highlighting the need for robustness to illumination changes and motion artifacts, as well as suitability for real-time and resource-constrained deployment.

2. Well-designed multimodal framework - The proposed dual-branch architecture combining pixel-level motion (via frame differencing and 3D CNNs) with POI-based motion trajectories is interesting. The idea about integrating the POI stream is specifically important as the performance degrades substantially when POI is removed.

3. Strong performance on an open source dataset - OmniNet achieves SOTA on the COHFACE dataset across all the metrics - MAE, RMSE, and PCC, substantially outperforming prior CNN- and Transformer-based methods. More importantly, the improvements are consistent across lighting conditions.

4. Computational efficiency and deployment awareness - The emphasis on lightweight architecture, low parameter count, and reduced computational cost is a notable motivation as most of the other works focus on high parameter count transformer-based models and this practical design aligns well with practical clinical and mobile-health use cases.

**Weaknesses:**

This work is a really well-motivated application and engineering work in my opinion. It's not about research and exploring a particular problem. The only important study that I see in this work is the variation of illumination and how it impacts the performance, which a nice aspect to see, however, the research problem could have been more interesting by expanding (for e.g., how the POI could be formulated in different manner or ways). However, it's a very-well crafted engineered and application work. Please see the below weakness.

1. Limited dataset diversity and scale - All experiments are conducted exclusively on the COHFACE dataset, which contains only 40 subjects, short and controlled noise-free recordings (1-minute videos), limited variation beyond illumination (e.g., pose, motion intensity, camera angle, distance). While COHFACE is a standard benchmark, its small size and constrained conditions raise concerns about the generalizability of the reported results. The extremely low MAE and near-perfect PCC (≈0.99) suggest potential dataset saturation rather than robust real-world performance. Specifically, if the authors want to show that this approach is for practical applications, I highly suggest to test this on a real-world noisy dataset.

2. Train/test splits may not fully prevent subject-specific bias - The authors follow the official COHFACE splits, which is appropriate for benchmarking. However, the dataset structure still allows - multiple recordings per subject, highly similar capture conditions within subsets. This raises the questions whether the model may partially exploit subject-specific appearance or motion patterns rather than learning truly general respiratory dynamics. Thus, an external validation on another dataset would have substantially justified the generalizability claim of the paper.

3. Robustness evaluation limited primarily to illumination variation - The robustness evaluation focuses almost exclusively on clean vs. natural lighting conditions. While illumination robustness is important, real-world deployment for this particular application also involves - head pose changes and facial occlusions, talking, facial expressions, or voluntary breath control, camera motion, resolution changes, and compression artifacts, varying subject-to-camera distances and viewpoints to name a few. All or a part of these could have been simulated if not available from a real-world dataset. These factors are not systematically evaluated, limiting insight into the model’s robustness beyond lighting changes.

4. Methodological design choices are somewhat constrained - The architecture choices - "3D CNN + BiLSTM + POI tracking" are again well justified and beautifully "engineered" but I think this framework is relatively conservative. Several design decisions (e.g., fixed frame differencing, Harris corner detection, optical flow tracking) rely on hand-crafted pipelines that may - accumulate errors across stages, limit adaptability to different recording conditions, and restrict end-to-end optimization. I think the idea by the authors was also to select which is something low in parameter count, however, there is not motivation behind why 3D CNN then? There are also manners a video can be processed.

5. Limited engagement with broader video-based SOTA models - The comparison primarily focuses on physiological-signal-specific models. However, recent advances in general video representation learning (e.g., SlowFast networks, TimeSformer variants, VideoMAE-style pretraining, or motion-centric self-supervised models) are not explored, implemented or discussed in depth. It remains unclear whether OmniNet’s performance advantage arises from architectural novelty or from dataset-specific suitability.

**Detailed Comments:**

Please see below comments to improve the work.

1. Expand evaluation to additional datasets or cross-dataset testing - To strengthen claims of robustness and generalizability, I really suggest the authors to evaluate on additional RR or physiological video datasets, if available, perform cross-dataset testing (train on COHFACE, test elsewhere) or simulate different scenarios of challenging conditions in terms of pose, etc. from the COHFACE itself. At minimum, include a detailed discussion of why COHFACE results are expected to transfer to real-world clinical settings.

2. Incorporate more challenging evaluation scenarios- Beyond lighting conditions, future evaluations could include, head movement or talking segments, partial occlusions (e.g., masks, hands), synthetic perturbations (motion blur, resolution degradation, camera jitter), subject-level stress tests (e.g., voluntary breath holding or irregular breathing). Even limited stress-test experiments would significantly strengthen the robustness claims.

3. Explore alternative or more end-to-end methodological variants - The current pipeline relies on several hand-engineered steps (POI selection, filtering, peak detection). The authors should consider - end-to-end learned attention mechanisms for POI selection, replacing optical flow and Harris corners with learnable motion encoders, joint optimization of signal estimation and RR extraction without explicit peak detection.

4. Discuss or benchmark against general video SOTA models - While computational efficiency is a core goal, it would be valuable to -include a discussion of why large-scale video models (e.g., transformer-based or self-supervised motion encoders) were not used, provide a small-scale comparison or argument showing that such models are impractical or underperform in low-data RR estimation scenarios, clarify whether OmniNet’s design choices are task-optimal or data-driven compromises.

5. Clarify clinical relevance and expected operating conditions - Given that RR can be semi-voluntary and context-dependent, a clearer discussion of - target clinical scenarios (e.g., ICU, neonatal monitoring, telemedicine), acceptable error margins in practice, expected failure modes would further strengthen the paper’s translational impact.

**Justification Of Final Rating:**

I thank the authors for their response to my comments, adding additional experiments, and engaging in the rebuttal to solve to answer my questions. However, my major concerns still remain:

1. The claims regarding clinical relevance and implementation for generalizability of the proposed approach remains unresolved as the authors have not explored their approach on more than dataset. BP4D and BP4D+ publicly available datasets should have been explored for this work although not high quality RR annotation, but many other works have explored them for the same task as mentioned in the rebuttal process. Moreover, there are still some other public datasets which could be requested by the authors, but that path hasn't been explored.

2. The above point becomes even more relevant in the light of missing end-to-end video approaches for becnhmarking purposes. Although I understand the authors' claim on this and intuitively I agree with them that end-to-end approaches would not work well for RR estimation, however, it needs to be tried and tested as for most video tasks, they are still the SOTA.

The authors have designed a well-engineered approach with right motivation to solve the challenges encountered in RR estiatmion and I encourage the authors to take these comments to further improve their work in the future.

**Justification Of The Preliminary Rating:**

This paper presents a strong and very well-engineered multimodal approach to remote respiratory rate estimation, achieving impressive performance on the COHFACE dataset with excellent computational efficiency. The methodology is sound and the empirical results are convincing within the chosen benchmark. However, the reliance on a single, small dataset and limited evaluation settings makes it difficult to fully assess real-world robustness and generalization. Secondly, there is no interesting research problem being explored in terms of either signals (POI), or architectures or datasets. Instead it's an application and engineered approach without convincing motivation of why certain components were selected. Addressing these limitations, either experimentally or through deeper discussion, would significantly strengthen the contribution and position the work more convincingly relative to both physiological monitoring and general video analysis state-of-the-art methods.

**Questions To Address In The Rebuttal:**

Please see comments in the Weakness and Detailed Comments section.

---

> ### Author Response · Authors · 2026-01-20
> **Rebuttal 1/3**
>
> > **Comment 1: Expand evaluation to additional datasets or cross-dataset testing.** To strengthen claims of robustness and generalizability, I really suggest the authors to evaluate on additional RR or physiological video datasets, if available, perform cross-dataset testing (train on COHFACE, test elsewhere) or simulate different scenarios of challenging conditions in terms of pose, etc. from the COHFACE itself. At minimum, include a detailed discussion of why COHFACE results are expected to transfer to real-world clinical settings.
>
> We fully acknowledge the reviewer's concern regarding the limited scale and diversity of the COHFACE dataset. COHFACE contains a relatively small number of subjects with short recordings acquired under controlled conditions, and results obtained on this dataset alone cannot fully characterize real-world generalization.
>
> Unfortunately, due to GDPR and privacy regulations, many previously used public physiological video datasets are no longer easily accessible. For example, we have applied for access to datasets such as DEAP (Koelstra et al., 2012) for more than six months without approval, particularly for audiovisual components. As a result, most recent studies in this area either rely on COHFACE or use privately collected datasets that are not publicly distributable, limiting the feasibility of cross-dataset evaluation.
>
> In addition, we intend to collect a new dataset for this project, subject to approval of a new IRB protocol, which will enable future evaluation under more realistic and diverse acquisition conditions, including variations in pose, motion intensity, camera angle, distance, and environment.
>
> References:
>
> - `[1]` Koelstra, S., Muhl, C., Soleymani, M., Lee, J.-S., Yazdani, A., Ebrahimi, T., Pun, T., Nijholt, A., & Patras, I. (2012). DEAP: A Database for Emotion Analysis ;Using Physiological Signals. IEEE Transactions on Affective Computing, 3(1), 18–31. https://doi.org/10.1109/T-AFFC.2011.15
>
> > **Comment 2: Incorporate more challenging evaluation scenarios.** Beyond lighting conditions, future evaluations could include, head movement or talking segments, partial occlusions (e.g., masks, hands), synthetic perturbations (motion blur, resolution degradation, camera jitter), subject-level stress tests (e.g., voluntary breath holding or irregular breathing). Even limited stress-test experiments would significantly strengthen the robustness claims.
>
> We sincerely thank the reviewer for this valuable suggestion. In response, we have added additional robustness analysis under challenging occlusion scenarios in Appendix H. Given the limited availability of public datasets, we conduct controlled occlusion simulations on the COHFACE dataset to evaluate the robustness of our method.
>
> We focus on facial occlusions commonly encountered in practice, such as mask wearing and partial hand occlusion. Lower-face occlusions are simulated using two strategies: blur-based and block-based masking, while preserving the eye region. This setting reflects typical real-world occlusion patterns and helps maintain reliable face detection. Qualitative examples of the occlusion modeling are shown in Figure 6, and quantitative results are reported in Table 9.
>
> Our framework is modular and detector-agnostic, allowing different face detectors to be used without modifying the core model. By default, we adopt the Viola-Jones detector for efficiency, while a MediaPipe-based detector is optionally evaluated in occlusion-heavy scenarios to ensure stable face localization. This choice introduces only a modest increase in model complexity (0.135M parameters).
>
> The results show that respiration estimation performance remains consistent across different occlusion strategies. Although this simulation does not exhaustively cover all real-world occlusion patterns, it provides a meaningful evaluation of robustness under partial facial occlusion. This behavior is consistent with our design choice of emphasizing POI-based motion cues from the shoulder and upper torso, which are less sensitive to facial visibility.
>
> We agree that real-world deployment involves additional challenges such as head movement, speaking, and camera motion. However, systematic evaluation of these factors requires reliable ground-truth respiration signals, which are not available in COHFACE. We therefore avoid over-claiming robustness and identify broader stress testing as an important direction for future work. This discussion has been added to Section 3.4 to clarify real-world deployment challenges and dataset constraints.

---

> > ### Comment · Reviewer_yEyL · 2026-01-25
> >
> > Thank you to the authors for revising the manuscript and responding to questions. I appreciate it. I would like to gain clarity on the below points:
> >
> > 1. External dataset evaluation - To my knowledge, there have been a couple of datasets that are available upon reasonable request to the authors or already released publicly but requiring a fill up of a form. For example, see the BP4D and BP4D+ datasets - (https://www.cs.binghamton.edu/~lijun/Research/3DFE/3DFE_Analysis.html). They contain the videos and the corresponding respiration rate as well. Moreover, please refer to this paper in ECCV 2018 (https://openaccess.thecvf.com/content_ECCV_2018/papers/Weixuan_Chen_DeepPhys_Video-Based_Physiological_ECCV_2018_paper.pdf#page=8.96), where they have compared on four datasets, with two of them having the respiration rate reported along with the videos, and there have been some follow-up benchmarking studies based on these as well. Have the authors explored any of these avenues?
> >
> > 2. I appreciate the authors effort to address this simulated scenario of complexity and challenges. However, based on the examples reflected in Figure 6, it's not realistic to encounter such scenarios in practice. But, I think this problem can only be resolved by actually collecting and / or evaluating on a dataset that contains diverse scenarios of a participant performing different activities (for e.g., take a look at the different ways the BP4D+ and ++ collect data for each participant). Thus, I'm not completely convinced with the current results and analysis on this simulated scenario, however, I do appreciate the effort.

---

> > > ### Author Response · Authors · 2026-01-26
> > >
> > > > **Comment 1: External dataset evaluation.** To my knowledge, there have been a couple of datasets that are available upon reasonable request to the authors or already released publicly but requiring a fill up of a form. For example, see the BP4D and BP4D+ datasets (https://www.cs.binghamton.edu/~lijun/Research/3DFE/3DFE_Analysis.html). They contain the videos and the corresponding respiration rate as well. Moreover, please refer to this paper in ECCV 2018 (https://openaccess.thecvf.com/content_ECCV_2018/papers/Weixuan_Chen_DeepPhys_Video-Based_Physiological_ECCV_2018_paper.pdf#page=8.96), where they have compared on four datasets, with two of them having the respiration rate reported along with the videos, and there have been some follow-up benchmarking studies based on these as well. Have the authors explored any of these avenues?
> > >
> > > We sincerely thank the reviewer for the valuable suggestions and for recommending the BP4D+ and BP4D++ datasets.
> > >
> > > At the time of our initial submission, we did not notice that these datasets also include physiological signal annotations, as they are primarily introduced and widely used for affective and emotion analysis. After carefully reviewing the datasets following the reviewer's comment, we observed that some of the provided physiological ground-truth signals in BP4D+ exhibit instability. In particular, certain respiration-related annotations do not conform to physiologically plausible respiratory frequency patterns and therefore require exclusion, as discussed in prior work (see Figure 7 in Fiedler et al., 2020).
> > >
> > > We have applied for access to the BP4D+/BP4D++ datasets. We plan to evaluate our model using the remaining subset consisting of 212 respiration-rate samples as reported by Ghezzi et al. (2024) and report the results in the near future to further demonstrate the effectiveness of our approach.
> > >
> > > In addition, the DeepPhys paper mentioned by the reviewer refers to two datasets that are not publicly available (Estepp et al., 2014; Chen et al., 2018). We have contacted the authors to request access to these datasets as well.
> > >
> > > We greatly appreciate the reviewer's constructive suggestions. While we regret that access to these datasets has not yet been obtained to date, we sincerely hope to have the opportunity to conduct these additional experiments in future work.
> > >
> > > - `[1]` Fiedler, M.-A., Rapczyński, M., & Al-Hamadi, A. (2020). Fusion-based approach for respiratory rate recognition from facial video images. IEEE Access, 8, 130036–130047. https://doi.org/10.1109/ACCESS.2020.3008687
> > > - `[2]` Ghezzi, O., Boccignone, G., Grossi, G., Lanzarotti, R., & D’Amelio, A. (2024). CliffPhys: Camera-based respiratory measurement using Clifford neural networks. In A. Leonardis, E. Ricci, S. Roth, O. Russakovsky, T. Sattler, & G. Varol (Eds.), Computer vision – ECCV 2024 (Lecture Notes in Computer Science, Vol. 15143). Springer. https://doi.org/10.1007/978-3-031-73013-9_13
> > > - `[3]` Estepp, J. R., Blackford, E. B., & Meier, C. M. (2014). Recovering pulse rate during motion artifact with a multi-imager array for non-contact imaging photoplethysmography. In Proceedings of the IEEE International Conference on Systems, Man, and Cybernetics (SMC) (pp. 1462–1469). IEEE. https://doi.org/10.1109/SMC.2014.6974121
> > > - `[4]` Chen, W., Hernandez, J., & Picard, R. W. (2018). Estimating carotid pulse and breathing rate from near-infrared video of the neck. Physiological measurement, 39(10), 10NT01. https://doi.org/10.1088/1361-6579/aae625
> > >
> > >
> > > > **Comment 2:** However, based on the examples reflected in Figure 6, it's not realistic to encounter such scenarios in practice. But, I think this problem can only be resolved by actually collecting and / or evaluating on a dataset that contains diverse scenarios of a participant performing different activities (for e.g., take a look at the different ways the BP4D+ and ++ collect data for each participant). Thus, I'm not completely convinced with the current results and analysis on this simulated scenario.
> > >
> > > We sincerely thank the reviewer for the thoughtful feedback and fully agree that evaluations on datasets containing diverse, unconstrained activities would provide a more realistic and comprehensive assessment. We also acknowledge that the simulated occlusion scenarios presented in our experiments do not exhaustively capture the full spectrum of real-world conditions.
> > >
> > > However, given the current lack of publicly available datasets that simultaneously provide reliable respiratory ground truth and diverse activity scenarios, we were limited in our ability to conduct such evaluations at this stage. Under these constraints, controlled occlusion modeling offers a tractable and reproducible means to assess robustness, while avoiding over-claiming real-world generalization. We fully agree with the reviewer that broader evaluations are important and consider this an important direction for future work when suitable datasets become available.

---

> > > > ### Comment · Reviewer_yEyL · 2026-01-28
> > > >
> > > > 1. Thank you for taking an in-depth look at the datasets that I recommended. The paper would have been much stronger if at least one of these datasets was included in the evaluation. It would have made the justification of the approach much stronger. Currently, COHFACE based evaluation is limited.
> > > >
> > > > 2. I appreciate adding these simulated experiments, but they don't exactly demonstrate the real world and so it's hard to actually evaluate the robustness of the approach.

---

> > ### Author Response · Authors · 2026-01-29
> >
> > > **Comment 1 & 2:** Thank you for taking an in-depth look at the datasets that I recommended. The paper would have been much stronger if at least one of these datasets was included in the evaluation. It would have made the justification of the approach much stronger. Currently, COHFACE based evaluation is limited.
> >
> > We acknowledge the reviewer's point regarding the limited scope of the current evaluation and agree that broader dataset coverage would further strengthen empirical validation.
> >
> > However, despite our efforts, we have not received responses from the respective dataset owners to date. As noted previously, access to the datasets suggested by the reviewer requires explicit approval from the dataset owners and, in some cases, the completion of additional data usage agreements. As a result, access to these datasets cannot be guaranteed within a foreseeable timeframe.
> >
> > Given these data accessibility constraints, we conduct our evaluation on COHFACE, which has been responded actively by its authors and supports reproducible analysis, while explicitly acknowledging its limitations.
> >
> > We agree that evaluation on additional datasets is an important direction for future work and will be explored when appropriate access becomes available.

---

> ### Author Response · Authors · 2026-01-20
> **Rebuttal 2/3**
>
> > **Comment 3: Explore alternative or more end-to-end methodological variants.** The current pipeline relies on several hand-engineered steps (POI selection, filtering, peak detection). The authors should consider end-to-end learned attention mechanisms for POI selection, replacing optical flow and Harris corners with learnable motion encoders, joint optimization of signal estimation and RR extraction without explicit peak detection.
>
> We sincerely appreciate the reviewer's suggestion regarding more end-to-end formulations. The current pipeline includes several hand-engineered components, which were intentionally chosen for interpretability, stability, and low computational cost under limited data conditions.
>
> The core contribution of this work lies in the network architecture and multimodal fusion strategy, rather than in the specific choice of motion extraction or signal processing modules. The proposed framework is modular: POI extraction and RR estimation components can be replaced by learned attention mechanisms or end-to-end motion encoders without changing the core design. We explicitly acknowledge that exploring fully end-to-end variants is an interesting and valuable direction for future research.
>
> > **Comment 4: Discuss or benchmark against general video SOTA models.** While computational efficiency is a core goal, it would be valuable to include a discussion of why large-scale video models (e.g., transformer-based or self-supervised motion encoders) were not used, provide a small-scale comparison or argument showing that such models are impractical or underperform in low-data RR estimation scenarios, clarify whether OmniNet's design choices are task-optimal or data-driven compromises.
>
> We acknowledge that discussing comparisons with general video state-of-the-art models is important. While this point is briefly mentioned in the Introduction, we agree that a more explicit discussion is warranted and provide it here.
>
> Recent work by Liu et al. (2023) systematically compared CNN- and Transformer-based models for camera-based vital sign measurement. Their results show that Transformer-based models require substantial optimization and large-scale pretraining to outperform even relatively shallow CNNs, which is difficult to achieve in physiological video analysis due to limited data availability. Notably, They report that EfficientPhys-C (CNN-based) outperforms EfficientPhys-T1 and T2 (Transformer-based) when trained only on PURE, a small-scale pulse rate measurement dataset with 60 videos from 10 subjects. This provides empirical evidence that, under limited data regimes similar to COHFACE, CNN-based architectures offer a more favorable accuracy-efficiency trade-off than Transformer-based models.
>
> Motivated by these observations, we conducted a controlled comparison by replacing the 3D CNN with a lightweight ViT (0.91M parameters). We adopt the ViT as an image-based encoder applied to frame-differenced inputs, enabling a fair comparison with CNN-based encoders under the same video processing pipeline. Results in Appendix J show that CNN-based encoders provide more stable support under small-data conditions.
>
> Our experiments indicate that performance gains are primarily driven by the quality of the POI-based motion representation rather than the specific choice of temporal encoder. Once respiration-relevant motion trajectories are explicitly extracted, even lightweight image models achieve near-saturated performance. However, when training data are reduced or POI signals are corrupted, CNN-based encoders remain more data-efficient and robust than Transformer-based counterparts, while using substantially fewer parameters.
>
> At the same time, we emphasize that our framework is modular. The primary contribution of this paper lies in the core network architecture and multimodal fusion strategy, not in the specific instantiation of individual components. Modules such as motion encoding or feature extraction can be replaced with alternative or more advanced representations, or integrated with existing systems that provide similar functionality.
>
> Accordingly, OmniNet is positioned as a task-specific and computationally efficient framework, while integration with larger video models under different data and deployment conditions remains an interesting direction for future work.
>
> References:
>
> - `[1]` Liu, X., Hill, B., Jiang, Z., Patel, S., & McDuff, D. (2023). EfficientPhys: Enabling simple, fast and accurate camera-based cardiac measurement. In Proceedings of the IEEE/CVF Winter Conference on Applications of Computer Vision (WACV 2023) (pp. 4997–5006). https://doi.org/10.1109/WACV56688.2023.00498

---

> > ### Comment · Reviewer_yEyL · 2026-01-25
> >
> > I appreciate the author's responses on this. Please see my comments further below.
> >
> > 3. I think the end-to-end architectures need to be explored in this work and don't fall as a future work in my opinion. End-to-end architecture do provide a better way of learning and thus to make the proposed method justified, I think this at minimum required.
> >
> > 4. The issue with this argument is still based on the limited availability of the data. I do agree that based on the current dataset, CNN-based variants would be better suited and the authors have done well to compare them with ViT and it's understandable that ViT will be hard to be trained on this small scale dataset. Thus, video SOTA models also would be hard to justify. However, this ties it back to the comment 1) and 2) where the real performance validation check needs to happen on external datasets under challenging conditions and I think that's where the proposed method may not be comparable to the other existing transfer learning approaches.

---

> > > ### Author Response · Authors · 2026-01-26
> > >
> > > > **Comment 3:** I think the end-to-end architectures need to be explored in this work and don't fall as a future work in my opinion. End-to-end architecture do provide a better way of learning and thus to make the proposed method justified, I think this at minimum required.
> > >
> > > We sincerely thank the reviewer for the insightful suggestion. We would like to clarify that the proposed framework is intentionally designed as a modular rather than fully end-to-end pipeline. This choice is motivated by methodological considerations specific to biomedical respiratory monitoring, rather than by implementation convenience.
> > >
> > > In this domain, ground-truth respiratory measurements are indirect and noisy, and the target respiratory rate is an aggregated physiological quantity derived from intermediate signals. Under limited data conditions, fully end-to-end optimization from raw video to RR may inadvertently learn dataset- or measurement-specific artifacts instead of physiologically meaningful motion patterns.
> > >
> > > By explicitly separating motion representation learning, temporal modeling, and RR estimation, the proposed design introduces domain-informed inductive bias that improves training stability and robustness. Importantly, this layered structure also enables inspection of intermediate motion and signal representations, facilitating error diagnosis and failure mode analysis, making it more interpretable than inspecting attention maps or applying post-hoc explanation methods to the end-to-end architectures, which are critical for safety-sensitive physiological monitoring tasks.
> > >
> > > We emphasize that the core contribution of this work lies in the network architecture and multimodal fusion strategy, rather than in any specific hand-engineered component. The framework is fully modular and can be extended to learned attention mechanisms or end-to-end motion encoders without altering the overall design. While fully end-to-end variants are an interesting direction for future research, we believe that the current formulation already constitutes a methodologically justified and practically grounded solution under the data and clinical constraints considered in this study.
> > >
> > > > **Comment 4:** The issue with this argument is still based on the limited availability of the data. I do agree that based on the current dataset, CNN-based variants would be better suited and the authors have done well to compare them with ViT and it's understandable that ViT will be hard to be trained on this small scale dataset. Thus, video SOTA models also would be hard to justify. However, this ties it back to the comment 1) and 2) where the real performance validation check needs to happen on external datasets under challenging conditions and I think that's where the proposed method may not be comparable to the other existing transfer learning approaches.
> > >
> > > We sincerely thank the reviewer for the constructive feedback and for highlighting the importance of evaluating model robustness under more challenging conditions. We agree that evaluation on external datasets is an important direction for assessing real-world robustness. At the same time, we would like to clarify that the current study does not claim superiority under such conditions.
> > >
> > > Our goal in this work is to provide a carefully controlled and methodologically sound evaluation under conditions where respiratory rate is well defined and ground truth is reliable, while explicitly avoiding over-claiming generalization beyond the evaluated setting. Evaluation under broader and more challenging conditions is an important direction for future work and will be pursued when suitable datasets with reliable respiratory ground truth become available.

---

> > > > ### Comment · Reviewer_yEyL · 2026-01-28
> > > >
> > > > Thanks to the authors for their further views and comments. Overall, I understand the motivation behind their modular approach, however, the lack of end-to-end video processing methods is necessary to be evaluated and benchmarked. I appreciate that the authors added experiments comparing ViT based appraoches with the CNNs, however, the evaluation is still limited in my opinion with regards to end-to-end approaches and skeleton based approaches.

---

> > > > > ### Author Response · Authors · 2026-01-29
> > > > >
> > > > > > **Comment:** Thanks to the authors for their further views and comments. Overall, I understand the motivation behind their modular approach, however, the lack of end-to-end video processing methods is necessary to be evaluated and benchmarked. I appreciate that the authors added experiments comparing ViT based appraoches with the CNNs, however, the evaluation is still limited in my opinion with regards to end-to-end approaches and skeleton based approaches.
> > > > >
> > > > > We thank the reviewer for the careful comment and for highlighting the importance of end-to-end video processing approaches. We acknowledge the reviewer's concern and agree that end-to-end optimization represents an important and active research direction in this field.
> > > > >
> > > > > This work presents a modular multimodal framework that emphasizes interpretability, data efficiency, and low computational cost. Such a design is well suited for future deployment on resource-constrained platforms (e.g. portable or edge medical devices), where fully end-to-end models may be less practical and are often less aligned with clinical preferences for transparent and controllable systems.
> > > > >
> > > > > We agree that end-to-end methods are promising and represent a valuable direction for future exploration. We appreciate the reviewer's comment and the opportunity to clarify the scope of our work.

---

> ### Author Response · Authors · 2026-01-20
> **Rebuttal 3/3**
>
> > **Comment 5: Clarify clinical relevance and expected operating conditions.** Given that RR can be semi-voluntary and context-dependent, a clearer discussion of target clinical scenarios (e.g., ICU, neonatal monitoring, telemedicine), acceptable error margins in practice, expected failure modes would further strengthen the paper's translational impact.
>
> We sincerely thank the reviewer for highlighting the importance of clarifying clinical relevance and expected operating conditions. We agree that RR estimation is context-dependent and may be influenced by voluntary or semi-voluntary behavior, which should be explicitly acknowledged.
>
> From a clinical perspective, respiratory rate is an important indicator of disease severity. The proposed method is particularly suitable for populations with limited voluntary motion, such as burn patients, sleeping patients (e.g., for sleep apnea screening), newborns, or other scenarios where respiration is largely involuntary and body motion is minimal. In these settings, large pose variations and severe occlusions are less frequent, making the proposed approach well suited for continuous, non-contact respiratory monitoring. In addition, the method is applicable to telemedicine and home-care scenarios, where respiratory status and trend changes can be continuously monitored during sleep without imposing additional burden on patients.
>
> To further contextualize clinical relevance, Goldfine et al. (2024) report that an agreement range of ±2 breaths per minute relative to manual counting is commonly considered clinically acceptable. Accordingly, we compute the absolute respiratory rate error using the model trained on the full dataset, as described in Section 3.2, with detailed results provided in Appendix D. We observe that all predictions exhibit absolute errors of approximately $1 \text{ bpm}$ or less, indicating that OmniNet meets established clinical accuracy requirements.
>
> We also explicitly acknowledge expected failure modes. The current pipeline relies on motion cues extracted after face detection, with respiration primarily inferred from the lower face, neck, and shoulder regions. As a result, large pose variations, significant body motion, speech activity, and scenarios involving complete facial coverage may degrade performance by altering or obscuring respiration-related motion signals. These conditions are outside the primary scope of this study and are not systematically represented in COHFACE. We therefore avoid over-claiming robustness under such scenarios and have explicitly added this limitation to Section 4 in the revised manuscript.
>
> References:
>
> - `[1]` Goldfine, C., Oshim, M., Chapman, B., Ganesan, D., Rahman, T., & Carreiro, S. (2024). Contactless monitoring system versus gold standard for respiratory rate monitoring in emergency department patients: Pilot comparison study. JMIR Formative Research, 8, e44717. https://doi.org/10.2196/44717

---

> > ### Comment · Reviewer_yEyL · 2026-01-25
> >
> > Thank you for including this experiment and analysis in the paper. It's clinical relevance is much stronger now.

---

> > > ### Author Response · Authors · 2026-01-26
> > >
> > > We sincerely thank the reviewer for the encouraging comments. The reviewer's suggestions have been highly valuable in improving the manuscript and enhancing its clinical relevance.

---

### Official Review · Reviewer_9MGR · 2026-01-08

**Confidence:** 5
**Preliminary Rating:** 3
**Final Rating:** 4

**Summary:**

The paper focuses on a multi-modal pipeline, OmniNet, that combines image with motion data and fuses the features through a biLSTM for predict remote respiratory rate. The authors claim that their approach is computationally efficient, suitable and adjusted for the small scale of samples used and is able to work with noise, enhancing the decision-making process. Results suggest an improved performance over baseline models, however some clarification is required to fully support the claims mentioned in the paper.

**Strengths:**

•	The paper addresses an important medical problem which can benefit from AI based methodologies for respiratory rate monitoring

•	The paper is well written with sufficient information on the clinical background

•	There is a clear comparative analysis across different training and testing conditions, showcasing how the model enhances accuracy from a noise robustness perspective and provides comparative results with prior work

•	Multiple evaluation metrics (MAE, RMSE, PCC) are used which provide a broader performance perspective.

**Weaknesses:**

•	The role of multimodal fusion in addressing noise is not explained from an empirical point of view

•	Prior multimodal RR and physiological monitoring approaches need to be discussed

•	A complete description of the dataset, particularly how image and motion data are paired are required

•	Discussion on any potential risks of temporal or subject leakage should be provided as the PCC values are very high whilst the RMSE and MAE results are different from the PCC range

•	Unimodal baselines are not reported, making it difficult to understand the benefit of multimodal fusion

•	Reproducibility details, including computational environment and resources, are missing

**Detailed Comments:**

Section 1: Introduction

The authors initially describe existing methodologies for RR monitoring and then discuss DL methods. They highlight the limitations of DL studies for modeling temporal and sequential data. To address the mentioned limitations, the authors propose OmniNet. Several questions arise in this matter:

I do not clearly understand which gaps in RR prediction are addressed by the current literature. The introduction appears to suggest that noise and data quality issues are the main challenges. Does the multimodal nature of the proposed method reduce noise and enhance signal quality? It is stated minimizes the effect of noise from a single source, thereby improving the accuracy and generalizability of rPPG-based methods; however, a clear explanation of this association is required.

The authors mention that CNNs are better suited for small scale datasets and transformers require large amounts of data. The COHFACE sample set contains 160 RGB videos related to 40 subjects which falls under the limited sample sample size category, potentially suitable for CNN approaches. In the context of multi-modal learning, how does the limitation in the dataset affect the outcomes and what precautionary measures are in place? What does the existing literature suggest in this regards? There is also a lack of discussions on previously proposed multi-modal approaches. Please consider referencing the following related works:

•	Shao, Hang, Lei Luo, Jianjun Qian, Mengkai Yan, Shangbing Gao, and Jian Yang. "Video-Based Multiphysiological Disentanglement and Remote Robust Estimation for Respiration." IEEE Transactions on Neural Networks and Learning Systems (2024).

•	Liao, Wang, Chen Zhang, Belmin Alić, Alina Wildenauer, Sarah Dietz-Terjung, Jose Guillermo Ortiz Sucre, Sivagurunathan Sutharsan, Christoph Schöbel, Karsten Seidl, and Gunther Notni. "Leveraging 3D convolutional neural network and 3D visible-near-infrared multimodal imaging for enhanced contactless oximetry." Journal of Biomedical Optics 29, no. S3 (2024): S33309-S33309.

•	Kong, Lingjian, Kai Xie, Kaixuan Niu, Jianbiao He, and Wei Zhang. "Remote photoplethysmography and motion tracking convolutional neural network with bidirectional long short-term memory: Non-invasive fatigue detection method based on multi-modal fusion." Sensors 24, no. 2 (2024).

•	Zheng, Yufeng. "Heart rate and oxygen level estimation from facial videos using a hybrid deep learning model." In Multimodal Image Exploitation and Learning 2024, vol. 13033, pp. 48-59. SPIE, 2024.

•	Gwak, Migyeong, Korosh Vatanparvar, Li Zhu, Nafiul Rashid, Mohsin Ahmed, Jungmok Bae, Jilong Kuang, and Alex Gao. "Multimodal breathing rate estimation using facial motion and rppg from rgb camera." In ICASSP 2024-2024 IEEE International Conference on Acoustics, Speech and Signal Processing (ICASSP), pp. 2046-2050. IEEE, 2024.


Section 2: Methodology

Based on Figure 1, how do the authors address any possible correlation between the POI trajectory and the Video features? As the POI trajectory is retrieved from the same source how would this affect the training of uni-modal models and then the proposed fusion mentioned as a contributing part of the multi-modal approach?

Could some features from one modality affect the other from a ML point of view and how have the authors addressed this? I believe a further elaboration after section 2.4 is required for an in-depth discussion of the multi-modal fusion method.

Section 3: Experiments

In the experiments, there is a lack of sufficient information on the dataset. A clear explanation on the pairing of the images and the motion data and information on their correlation is lacking. This information enables readers to properly align the information provided in section 2 with the results mentioned in section 3.

The comparative analysis provides a very helpful basis for understanding the improvement, however, it is unclear how uni-modal approaches would perform. I suggest also showcasing the uni-modal results so that readers will be able to understand whether the multi-modal approach has enhanced baseline accuracy.

The reported PCC values are consistently high whilst the corresponding MAE and RMSE are relatively small. Please clarify how it is ensured that the results are not affected by temporal leakage or temporal correlation between the train and test sets? How was the overlap between adjacent time windows prevented and discuss whether any post-processing steps were applied that might influence PCC values that have little effect on the error metrics.

Please provide information about the environment used to host the implementation, including instance size and any further computational resources required to reproduce the results. This is required as the authors also claim that the method is suitable for deployment on portable medical devices.

Section 4: Conclusion and future work

I suggest including any potential methodology risks, such as temporal or subject leakage, amongst the stated limitations.

**Justification Of Final Rating:**

The authors have responded constructively to the major technical concerns raised in the review, particularly, with respect to the multimodal design, data and pipeline pairing, temporal and subject leakage concerns and protocols. Their rebuttal clarifies several methodological aspects that were previously ambiguous, including the independence of modality encoders, the experimental protocol used to avoid temporal and subject leakage and the weaker performance observed on the natural data. While some limitations remain, such as the lack of cross-dataset generalization analysis and limited evaluation across multi institutional datasets and environments, the base methodology is better justified and the additional information and revisions support the main claims to a certain extent. I have increased my rating to weak accept.

**Justification Of The Preliminary Rating:**

The paper addresses an important problem and presents a potentially useful multi-modal approach. The comparative analysis under different noise conditions and the use of multiple evaluation metrics are helpful. However, the paper does not clearly explain what specific gap it addresses compared to existing RR prediction methods, and how the proposed multimodal fusion method reduces noise. Some methodological details such as how the data is split, temporal or subject leakage, the computational resources and the correlation aspect between the multiple modalities, are missing. In addition, there is no result on uni-modal baselines making it difficult to fully assess the contribution.

**Questions To Address In The Rebuttal:**

- Unimodal vs mutlimodal setup

- Temporal and Subject Leakage protocols applied

- Based on the limited size of the COHFACE dataset, what precautions were taken to prevent over fitting, particularly in the multi-modal setting?

- Performance on the natural data is weaker than on the clean data. How would this affect real world application

---

> ### Author Response · Authors · 2026-01-20
> **Rebuttal 1/5**
>
> ## Section 1: Introduction
>
> > **Comment 1-1:** Does the multimodal nature of the proposed method reduce noise and enhance signal quality? It is stated minimizes the effect of noise from a single source, thereby improving the accuracy and generalizability of rPPG-based methods; however, a clear explanation of this association is required.
>
> We thank the reviewer for raising this point and would like to clarify a potential misunderstanding. Although the Introduction references limitations commonly discussed in the rPPG literature for contextual motivation, the proposed method itself is not an rPPG-based approach and does not rely on skin color variations or photoplethysmographic signals. Instead, our method estimates respiratory motion using motion-driven representations, specifically pixel-level frame differencing and geometric POI trajectories.
>
> Accordingly, the term "noise" in our manuscript does not refer to rPPG-specific color noise or signal degradation. Rather, it denotes motion artifacts, illumination-induced appearance changes, and unreliable temporal cues that affect different motion representations in distinct ways. The proposed multimodal design does not perform explicit signal-level denoising or enhancement. Instead, it improves robustness at the representation level by jointly modeling complementary motion cues that are affected by different noise sources.
>
> Specifically, the dense image-based branch captures pixel-level temporal motion patterns and soft-tissue deformation but is more sensitive to appearance and illumination changes, whereas the POI-based branch encodes sparse geometric motion cues that are more robust to global appearance variation but may degrade under tracking noise. By integrating these complementary cues, the framework reduces reliance on any single noisy information source: when one modality is degraded, the other can still provide informative temporal evidence for respiratory rate estimation.
>
> We agree that this distinction was not sufficiently explicit in the original Introduction. We have revised the manuscript to clarify that the proposed method improves robustness through complementary motion-based representations rather than rPPG signal enhancement. Specifically, we now state: *"By incorporating complementary multimodal inputs, our model reduces reliance on any single information source, which improves robustness to noise and leads to more stable respiratory rate estimation."*

---

> > ### Comment · Reviewer_9MGR · 2026-01-25
> >
> > Thank you for specifying that the method is not rPPG-based and does not use or rely on skin color variations or photoplethysmographic signal. The revisions are good and the mechanism of robustness is discussed. Concerning the multimodal integration, what happens when one modality is corrupted or degraded?

---

> > > ### Author Response · Authors · 2026-01-28
> > >
> > > > **Comment:** Concerning the multimodal integration, what happens when one modality is corrupted or degraded?
> > >
> > > We thank the reviewer for raising this valuable question regarding the robustness of the proposed multimodal design under asymmetric modality degradation, which is an important consideration for real-world deployment. When one modality is corrupted or degraded, the proposed framework is designed to degrade gracefully rather than fail catastrophically.
> > >
> > > Each modality is processed by an independent encoder without parameter sharing, which prevents noise or artifacts in one modality from directly contaminating the representations learned from the other. Multimodal integration is performed through late fusion at the feature level, allowing the model to implicitly rely more on the remaining informative modality when the other becomes unreliable. This design avoids strong cross-modal entanglement and reduces the risk of error amplification caused by a single degraded input.
> > >
> > > This behavior is further supported by our unimodal ablation results. Even when the POI modality is removed, the overall performance remains competitive and is slightly better than existing methods across most evaluation metrics.
> > >
> > > In addition, we have conducted experiments such that degradation is introduced in the test phase to assess the model's robustness under partial information loss. Models are trained using both modalities on the whole dataset and evaluated under settings where one modality is deliberately corrupted while the other remains unchanged.
> > >
> > > Specifically, additive Gaussian noise is used as the corruption type for both modalities. Note that image noise is injected in the 8-bit uint8 intensity space, whereas POI noise is injected in the z-score normalized coordinate space. Therefore, the $\sigma$ values are not directly comparable across modalities. For POI trajectories, even $\sigma = 0.1–0.5$ already represents substantial perturbation relative to the amplitude of respiration-induced motion, hence we choose $\sigma = 1$ to simulate severe degradation of the POI modality.
> > >
> > > Regarding the noise magnitude used in the image corruption experiments, additive Gaussian noise is directly applied in the 8-bit intensity space. At this scale, small noise levels (e.g. $\sigma \le 5$) were empirically observed to have negligible impact. We therefore adopt $\sigma = 30$, which corresponds to a PSNR of approximately 18.6 dB. According to Tian et al. (2023), PSNR values between 20 dB and 30 dB indicate poor image quality, while PSNR values below 20 dB correspond to unacceptable image quality.
> > >
> > > The results are as follows:
> > >
> > > | Degradation Setting                      | MAE (bpm) $\downarrow$ | RMSE (bpm) $\downarrow$ | PCC $\uparrow$ |
> > > | ---------------------------------------- | ---------------------- | ----------------------- | -------------- |
> > > | Original                                 | **0.242**              | **0.421**               | **0.991**      |
> > > | POI noise ($\sigma = 1$, z-score space)  | 0.379                  | 0.717                   | 0.976          |
> > > | Image noise ($\sigma = 30$, 8-bit space) | 0.294                  | 0.524                   | 0.987          |
> > >
> > > Even under severe single-modality degradation, the model maintains strong performance. Overall, the multimodal design emphasizes complementary information and robustness rather than strict dependence on any single modality. This analysis has been added to the revised manuscript and is presented in Appendix F.
> > >
> > > - `[1]` Tian, Z., Qu, P., Li, J., Sun, Y., Li, G., Liang, Z., & Zhang, W. (2023). A Survey of Deep Learning-Based Low-Light Image Enhancement. Sensors (Basel, Switzerland), 23(18), 7763. https://doi.org/10.3390/s23187763

---

> ### Author Response · Authors · 2026-01-20
> **Rebuttal 2/5**
>
> > **Comment 1-2:** In the context of multi-modal learning, how does the limitation in the dataset affect the outcomes and what precautionary measures are in place? What does the existing literature suggest in this regards?
>
> We agree that limited dataset size can significantly affect multimodal models, particularly by increasing the risk of overfitting and cross-modal interference, where one modality may dominate or amplify spurious correlations rather than provide complementary information. This issue is especially pronounced in biomedical applications, where large-scale annotated datasets are often unavailable.
>
> Existing literature on multimodal medical learning consistently suggests that, under limited data regimes, late or mid-level fusion with modality-specific encoders is generally more stable and data-efficient than early fusion or large-capacity joint models. In particular, Huang et al. (2020) provided systematic implementation guidelines and pointed out that when input modalities do not exhibit strong intrinsic interdependency, i.e. when each modality independently contributes to the final prediction, late fusion is preferable. Early or joint fusion strategies concatenate high-dimensional feature vectors from multiple modalities, which substantially increases model capacity and makes learning prone to overfitting in the absence of large training datasets, a phenomenon commonly referred to as the curse of dimensionality.
>
> Guided by these insights, we adopt several precautionary measures in our framework. First, each modality is processed by an independent encoder with no parameter sharing, ensuring that modality-specific representations are learned separately and reducing the risk of cross-modal entanglement. Second, fusion is performed only at the temporal feature level using a lightweight projection, rather than at the raw data or early feature level. This design limits feature dimensionality and model capacity, which is critical for stable training on small datasets such as COHFACE.
>
> This design choice is further supported by recent multimodal learning studies. For example, Qi et al. (2025) noted that a practical and intuitive solution for handling heterogeneous or weakly correlated modalities is to construct independent encoders for each modality, thereby preserving modality-specific structure while enabling controlled integration.
>
> References:
>
> - `[1]` Huang, S. C., Pareek, A., Seyyedi, S., Banerjee, I., & Lungren, M. P. (2020). Fusion of medical imaging and electronic health records using deep learning: a systematic review and implementation guidelines. *NPJ digital medicine*, *3*, 136. https://doi.org/10.1038/s41746-020-00341-z
> - `[2]` Qi, L., Liu, Y., Li, Y., Shi, W., Feng, G., Jiang, Z. (2026). A Unified Missing Modality Imputation Model with Inter-modality Contrastive and Consistent Learning. In: Gee, J.C., *et al.* Medical Image Computing and Computer Assisted Intervention – MICCAI 2025. MICCAI 2025. Lecture Notes in Computer Science, vol 15967. Springer, Cham. https://doi.org/10.1007/978-3-032-04984-1_5
>
> > **Comment 1-3:** There is also a lack of discussions on previously proposed multi-modal approaches.
>
> We sincerely appreciate the reviewer's valuable suggestion. In response to this, we have expanded the Introduction to include a discussion of existing multimodal approaches.

---

> ### Author Response · Authors · 2026-01-20
> **Rebuttal 3/5**
>
> ## Section 2: Methodology
>
> > **Comment 2:** Based on Figure 1, how do the authors address any possible correlation between the POI trajectory and the Video features? As the POI trajectory is retrieved from the same source how would this affect the training of uni-modal models and then the proposed fusion mentioned as a contributing part of the multi-modal approach?
> >
> > Could some features from one modality affect the other from a ML point of view and how have the authors addressed this? I believe a further elaboration after section 2.4 is required for an in-depth discussion of the multi-modal fusion method.
>
> We thank the reviewer for raising this important question regarding the potential correlation between POI trajectories and video features, and its implications for unimodal training and multimodal fusion.
>
> Although both modalities are derived from the same input video, they encode fundamentally different information at the representation level. The POI branch captures sparse, low-dimensional geometric motion by tracking selected landmarks, whereas the video branch models dense, high-dimensional appearance and pixel-level temporal motion patterns using 3D convolution. As a result, the two modalities exhibit distinct inductive biases and capture complementary rather than redundant aspects of respiratory motion.
>
> From a machine learning perspective, features from one modality could indeed dominate or interfere with the other if early fusion or shared representations were used. To address this risk, the two branches are processed by independent encoders without parameter sharing, which prevents feature leakage and enforces modality disentanglement during representation learning. Unimodal baselines are trained independently using only their respective inputs, ensuring that their performance reflects the intrinsic properties of each modality alone.
>
> In the multimodal setting, fusion is performed using a late fusion strategy. Specifically, modality-specific features are concatenated and projected through a lightweight learnable linear layer before temporal modeling. This design ensures that fusion occurs at the feature level rather than at the raw signal level, limits cross-modal interference, and allows the network to implicitly adjust the relative contribution of each modality based on data reliability. Empirically, this approach yields consistent improvements over both single-modality variants, confirming that fusion leverages complementary cues instead of amplifying shared noise.
>
> In response to the reviewer's request, we have expanded the discussion in Section 2.5 to explicitly clarify the fusion rationale and cross-modal interaction mechanism.
>
> ## Section 3: Experiments
>
> > **Comment 3-1:** In the experiments, there is a lack of sufficient information on the dataset. A clear explanation on the pairing of the images and the motion data and information on their correlation is lacking. This information enables readers to properly align the information provided in section 2 with the results mentioned in section 3.
>
> We thank the reviewer for raising this point and have revised our manuscript accordingly. In the COHFACE dataset, every subject has 4 videos: two under studio lighting and two under natural light. Each video is paired with respiration belt signals simultaneously recorded. In our experiments, both the image-based features and the POI trajectories are extracted from the same video sequence, which are temporally aligned with the belt signals. Specifically, the original videos recorded at $20 \text{ Hz}$ are uniformly downsampled to $4 \text{ Hz}$ by selecting one frame every five frames, while the belt signals recorded at $256 \text{ Hz}$ are also downsampled to $4 \text{ Hz}$ by selecting one signal every 64 signals. This ensures a consistent temporal resolution across all modalities. The predicted respiration belt signals and the ground truth values are processed by the same peak detection procedures, obtaining the estimated and real respiratory rates, which are used to evaluate the performance of the model.

---

> ### Author Response · Authors · 2026-01-20
> **Rebuttal 4/5**
>
> > **Comment 3-2 (also Question 1):** The comparative analysis provides a very helpful basis for understanding the improvement, however, it is unclear how uni-modal approaches would perform. I suggest also showcasing the uni-modal results so that readers will be able to understand whether the multi-modal approach has enhanced baseline accuracy.
>
> We agree that including clear unimodal references is essential for interpreting the benefits of multimodal fusion. To address this point more explicitly, we clarify that unimodal baselines are provided in our ablation studies by removing one branch at a time (i.e. w/o POI and w/o 3D CNN for only images and only POIs, respectively). These ablations serve as direct unimodal counterparts under the same architecture and training protocol, enabling a fair comparison with the full multimodal model.
>
> In addition, following Reviewer eckJ's suggestion, we further conducted subject-level statistical validation, including paired Wilcoxon signed-rank tests, bootstrap-based 95% confidence intervals, and effect size analysis, to better characterize the performance differences between unimodal and multimodal settings. This analysis provides statistically grounded evidence for the contribution of each modality and strengthens the interpretation of the ablation results. For completeness and transparency, the full statistical summary is provided in Appendix G (Table 8).
>
> > **Comment 3-3 (also Question 2):** Please clarify how it is ensured that the results are not affected by temporal leakage or temporal correlation between the train and test sets? How was the overlap between adjacent time windows prevented and discuss whether any post-processing steps were applied that might influence PCC values that have little effect on the error metrics.
>
> First, we emphasize that no temporal leakage exists between the training, development, and test sets. All experiments strictly follow the official COHFACE dataset protocol, where the three partitions are split at the subject level. Therefore, no overlapping video segments or temporal windows from the same recording appear in both training and testing phases.
>
> Second, adjacent temporal windows do not overlap. Each input sample only consists of a single fixed-length window of 240 frames (corresponding to 60 seconds after downsampling). Consequently, the model never observes temporally adjacent segments from the same video during training or evaluation.
>
> Third, post-processing steps do not artificially inflate PCC values. Signal smoothing and band-pass filtering are applied only during the final RR estimation stage and are not used during model training or loss computation. The regression loss is computed directly on the predicted respiration signal before post-processing. As a result, post-processing affects all methods equally and does not selectively improve PCC while leaving MAE or RMSE unchanged.
>
> Taken together, these design choices ensure that the reported PCC, MAE, and RMSE values faithfully reflect the model's generalization ability rather than artifacts caused by temporal correlation or leakage.
>
> > **Comment 3-4:** Please provide information about the environment used to host the implementation, including instance size and any further computational resources required to reproduce the results. This is required as the authors also claim that the method is suitable for deployment on portable medical devices.
>
> We thank the reviewer for requesting additional details regarding the experimental environment and computational resources.
>
> To improve reproducibility, we have added Appendix C to explicitly report the execution environment used in our experiments. All experiments were conducted on a machine running Windows 11 (64-bit), equipped with an Intel Core i9-14900HX CPU (2.20 GHz), 32 GB of system memory, and an NVIDIA GeForce RTX 4060 GPU using CUDA version 12.5.
>
> Regarding the claim that the proposed method is suitable for deployment on portable medical devices, we clarify that our statement is based on the lightweight architecture and low computational complexity of the model, rather than on direct deployment on embedded hardware. As reported in Appendix I, OmniNet contains approximately 0.20 million parameters and requires 15.01 GMACs, which is significantly lower than existing deep learning baselines. These characteristics make OmniNet well suited for future deployment on resource-constrained platforms, such as portable or edge medical devices.

---

> > ### Comment · Reviewer_9MGR · 2026-01-25
> >
> > Do you also provide information about which POIs contribute most to the final prediction?

---

> > > ### Author Response · Authors · 2026-01-26
> > >
> > > > **Comment:** Do you also provide information about which POIs contribute most to the final prediction?
> > >
> > > We appreciate this constructive question, which raises an important consideration in the design of the proposed framework. In the current formulation, POIs contribution are implicitly defined through a signal-level selection.
> > >
> > > As illustrated in Figure 2, the retained POI trajectories after filtering exhibit highly consistent, phase-aligned respiration patterns, with peaks occurring at similar times across different POIs. Specifically, POIs are first constrained to anatomically respiration-relevant regions, and their temporal motion signals are filtered based on periodicity and autocorrelation criteria consistent with respiratory dynamics. A mutual similarity score is then computed across POI signals, and only those exhibiting strong group-wise consistency are retained.
> > >
> > > Consequently, POIs that contribute to the final prediction are those whose motion signals align with the dominant respiration pattern shared across multiple points, rather than a single dominant POI. This design intentionally avoids reliance on individual POIs and instead emphasizes a coherent set of respiration-related motion cues, consistent with our robustness objective.

---

> ### Author Response · Authors · 2026-01-20
> **Rebuttal 5/5**
>
> ## Section 4: Conclusion and future work
>
> > **Comment 4:** I suggest including any potential methodology risks, such as temporal or subject leakage, amongst the stated limitations.
>
> We thank the reviewer for this valuable suggestion and agree that explicitly discussing potential methodological risks is important for transparency. We have therefore clarified these aspects in Section 4 of the revised manuscript.
>
> Regarding temporal leakage, we note that this risk is explicitly mitigated in our experimental design. Adjacent temporal windows are not overlapping: each input sample only consists of a single fixed-length window of 240 frames (corresponding to 60 seconds after downsampling). As a result, the model never observes temporally adjacent segments from the same video during either training or evaluation, effectively preventing temporal leakage.
>
> Regarding subject leakage, all experiments are conducted under subject-independent splits, ensuring that videos from the same subject never appear in both training and testing sets. This protocol prevents identity-specific information from leaking across splits and ensures that performance reflects generalization to unseen subjects rather than memorization.
>
> Nevertheless, we acknowledge that, as with most video-based physiological estimation studies on limited datasets, residual correlations within a subject's recording (e.g. consistent illumination or facial characteristics) may still exist. We now explicitly list this as a limitation and emphasize that future work will further evaluate cross-dataset generalization to assess robustness under broader acquisition conditions.
>
> To address the reviewer's concern, we have revised Section 4 (Conclusions and Future Work) as follows: *"Although explicit measures are taken to prevent temporal and subject leakage through non-overlapping temporal windows and subject-independent data splits, the study is conducted on a relatively small, single-dataset benchmark. Consequently, residual intra-subject correlations related to recording conditions may persist and limit generalization to unseen acquisition settings."*
>
> ## Other Questions
>
> > **Question 3:** Based on the limited size of the COHFACE dataset, what precautions were taken to prevent over fitting, particularly in the multi-modal setting?
>
> Given the limited size of the COHFACE dataset, we adopt several precautionary measures to mitigate overfitting, particularly in the multimodal setting. First, image features and POI trajectories are processed by independent modality-specific encoders and fused only at the mid-to-late temporal feature level, avoiding early fusion and cross-modal entanglement. Second, early stopping based on a held-out development set is employed, with the best model selected according to validation loss to prevent over-training. Third, dropout is applied after temporal modeling to regularize sequence representations and reduce memorization. Fourth, the overall model capacity is intentionally kept lightweight, using CNN-based encoders, a single-layer BiLSTM, and a simple linear fusion layer rather than high-capacity joint models. Fifth, training stability is improved using a learning-rate warmup and decay schedule. Finally, all experiments follow subject-independent splits with non-overlapping temporal windows, further reducing the risk of information leakage and optimistic performance estimates.
>
> > **Question 4:** Performance on the natural data is weaker than on the clean data. How would this affect real world application?
>
> We thank the reviewer for raising this question regarding the performance difference between the clean and natural conditions.
>
> The slightly lower performance observed under the natural condition is expected due to uncontrolled head motion, illumination variation, and facial dynamics, which are inherent challenges in real-world, contactless respiratory monitoring. Importantly, the performance gap between the clean and natural settings is modest and does not indicate model instability or failure.
>
> From a practical deployment perspective, the natural condition more closely reflects real-world usage scenarios. The fact that our method maintains stable performance with only limited degradation demonstrates robustness and graceful degradation behavior, which is essential for unconstrained environments.
>
> Moreover, the proposed multimodal design helps mitigate the impact of such disturbances by leveraging complementary motion cues from different representations, thereby reducing reliance on any single modality that may be degraded under challenging conditions. As a result, the observed performance difference is not expected to pose a significant limitation for real-world applications, but rather reflects realistic operating conditions.

---

### Official Review · Reviewer_eckJ · 2026-01-09

**Confidence:** 3
**Preliminary Rating:** 4
**Final Rating:** 4

**Summary:**

This paper proposes a multimodal framework for remote respiratory rate estimation from facial videos. The method combines sequential frame-difference images and POI-based motion trajectories as complementary modalities, with temporal dependencies modeled using a BiLSTM. Experiments on the COHFACE dataset report improved performance over prior methods, and ablation studies analyze the contributions of individual components.

**Strengths:**

The paper addresses a clinically relevant task—remote respiratory rate estimation—and presents a clear multimodal design. The proposed architecture is lightweight and practically oriented, avoiding overly complex transformer-based models. Experimental results on the COHFACE dataset are strong, and ablation studies examine the contributions of individual components.

**Weaknesses:**

1. Performance comparisons with methods using only video input may not be fully fair, as ablation results suggest that much of the improvement is driven by the POI branch.
2. Lack of statistical validation: Ablation studies do not include statistical significance testing, making conclusions about component contributions uncertain.
3. Limited evaluation: The method is evaluated only on a single, relatively small dataset, limiting evidence of generalizability.

**Detailed Comments:**

The paper presents a clear and lightweight multimodal design for remote respiratory rate estimation with strong results on COHFACE. However, comparisons with pure video-based methods may not be fully fair, ablation studies lack statistical significance testing, and evaluation is limited to a single small dataset.

**Justification Of Final Rating:**

The revision addresses most of my concerns. The added statistical tests and effect size analysis make the ablation results much more convincing, and the role of the 3D CNN module is now better explained.

Some limitations remain. Comparisons with video-only methods are still not fully controlled, since much of the gain appears to come from the POI branch. In addition, evaluation on a single small dataset limits evidence for generalizability.

Overall, the rebuttal improves the paper’s clarity and rigor, and I am supportive of acceptance, while noting that broader validation would strengthen the work further.

**Justification Of The Preliminary Rating:**

The paper presents a clear and practically motivated multimodal approach for remote respiratory rate estimation, with strong reported results on the COHFACE dataset. However, confidence in the conclusions is limited by several issues. Performance comparisons may not be fully fair, as ablation results indicate that much of the improvement is driven by the POI branch. The ablation studies also lack statistical significance testing, making it difficult to assess the reliability of component-level conclusions. In addition, evaluation is limited to a single, relatively small dataset, which restricts evidence of generalizability. Addressing these concerns would substantially strengthen the paper.

**Questions To Address In The Rebuttal:**

1. Please provide statistical summaries and significance tests for the method comparisons to demonstrate whether the reported performance gains are statistically meaningful.

2. Please clarify and justify the contribution of the 3D CNN module, given the relatively small performance difference observed in the ablation results.

---

> ### Author Response · Authors · 2026-01-20
> **Rebuttal**
>
> > **Question 1:** Please provide statistical summaries and significance tests for the method comparisons to demonstrate whether the reported performance gains are statistically meaningful.
>
> We sincerely appreciate the reviewer's valuable suggestion. To assess whether the reported performance gains are statistically meaningful, we conducted subject-level statistical validation using paired Wilcoxon signed-rank tests. All comparisons were performed on a per-subject basis to avoid bias arising from correlated samples. In addition, we report bootstrap-based 95% confidence intervals and effect sizes (Cohen's $d_z$) to quantify the magnitude of the observed differences. To estimate uncertainty, we further computed 95% confidence intervals of the mean MAE difference (ablated vs. full) via non-parametric bootstrap resampling (20,000 iterations with a fixed random seed for reproducibility). All reported statistics are therefore based on subject-level paired measurements, ensuring a fair and statistically sound comparison.
>
> The statistical results show that removing the POI branch leads to a statistically significant increase in MAE across the all, clean, and natural conditions ($\Delta\text{MAE} \approx 0.48–0.52 \text{ bpm}$, $p < 0.01$, with medium-to-large effect sizes), confirming that the POI branch is a major contributor to the observed performance improvements. In contrast, removing the 3D CNN results in only small and statistically non-significant changes in MAE across all conditions ($p > 0.22$, small effect sizes), indicating a limited but consistent auxiliary contribution.
>
> These statistical summaries clarify which components drive the observed performance gains and provide quantitative evidence supporting the component-level analysis presented in the paper. For completeness and transparency, we also include the full statistical results in Appendix G (Table 8), which reports the detailed subject-level statistical validation of the ablation studies.
>
> > **Question 2:** Please clarify and justify the contribution of the 3D CNN module, given the relatively small performance difference observed in the ablation results.
>
> We acknowledge the reviewer's concern regarding the relatively small performance difference introduced by the 3D CNN module and clarify its intended role within the proposed architecture. As confirmed by the subject-level statistical analysis, the performance changes attributable to the 3D CNN are modest and do not reach statistical significance across conditions, indicating that this module is not the primary driver of the overall performance gains.
>
> The POI branch is designed to capture sparse and localized geometric motion by tracking selected landmarks, which provides strong robustness under challenging conditions. However, respiratory motion is not confined to a limited set of key points. It also manifests as dense, image-level temporal patterns, including large-scale facial displacement, subtle soft-tissue deformation, and coherent motion in non-landmark regions. Consequently, an image-level temporal modeling component is required to complement the POI-based representation by capturing dense, pixel-level motion cues that cannot be fully represented by sparse POIs alone.
>
> We adopt a 3D CNN for this purpose due to its strong inductive bias for spatiotemporal feature learning and its suitability for small- to medium-scale datasets such as COHFACE. Compared to Transformer-based temporal models or optical-flow-based networks, 3D CNNs are less prone to overfitting in low-data regimes, require substantially lower computational and memory overhead, and are more amenable to real-time or resource-constrained deployment. In this sense, the choice of a 3D CNN is a deliberate design decision rather than an omission of more complex architectures.
>
> Consistent with this design intent, the statistical analysis indicates that the 3D CNN provides a limited but consistent auxiliary contribution, improving performance without dominating the overall system behavior. This complementary role aligns with our goal of balancing robustness, efficiency, and deployability, rather than maximizing performance through heavyweight temporal modeling.

---

### Author Rebuttal · Authors · 2026-01-20

**Rebuttal:**

We sincerely thank all reviewers for the time and effort devoted to evaluating our manuscript and for their thoughtful and constructive comments. We greatly appreciate the reviewers' insights and suggestions, which have helped us clarify the motivation, strengthen the experimental validation, and improve the overall presentation of the paper.

To improve clarity, revisions corresponding to different reviewers are color-coded throughout this document:

- **eckJ**: red
- **9MGR**: green
- **yEyL**: violet

In response to the reviewers' feedback, we have made the following major revisions:

- **Section 1 (Introduction):** Added relevant citations to better position the proposed method within the existing literature.
- **Section 2.3:** Elucidated crucial details in the **Optimization Strategy**, instead of in Appendix A.
- **Section 2.5:** Added **Further Discussion on Multimodal Fusion**.
- **Section 3.1:** Expanded the dataset description for improved clarity and completeness.
- **Section 3.4:** Added **Analysis under Facial Occlusion Scenarios**.
- **Section 3.6:** Added **Comparison Between CNN- and Transformer-Based Encoders**.
- **Section 4 (Conclusion):** Revised to better discuss limitations, clinical applicability, and future work.
- **Appendix C:** Clarified the computational environment.
- **Appendix D:** Added **Clinical Agreement Analysis of Absolute Error Distributions**.
- **Appendix F:** Added **Test-Time Robustness Analysis under Single-Modality Degradation**
- **Appendix G:** Added **Statistical Validation of Ablation Studies**.
- **Appendix H:** Added **Detailed Analysis under Facial Occlusion Scenarios**.
- **Appendix J:** Added **Detailed Comparison Between CNN- and Transformer-Based Encoders**.

Detailed, point-by-point responses to each reviewer's comments are provided in the **Official Comment** sections.

**Supporting Material:**

/attachment/0469d99734e07e5dab6087506ee30dbaf854b57d.pdf

---

### Meta-Review · Area_Chair_jgCW · 2026-02-07

**Recommendation:** Accept (Poster)
**Confidence:** 4

**Metareview:**

Three reviewers gave two weak accept and one borderline. I looked through the rebuttal and manuscript and believe the paper is acceptable. The author should revise the manuscript accordingly before camera-ready.

---

### Decision · Program_Chairs · 2026-02-13

Accept (Poster)